# HALLUCINATION REDUCTION WITH CASAL: CONTRASTIVE ACTIVATION STEERING FOR AMORTIZED LEARNING

Wannan (Winnie) Yang[*,1,2], Xinchi Qiu[3], Lei (Jade) Yu[1], Yuchen Zhang[1], Aobo Yang[3],
Narine Kokhlikyan[1], Nicola Cancedda[1], Diego Garcia-Olano[1]
[1]Meta Superintelligence Labs    [2]New York University    [3]Meta
[*]Work done at Meta

## ABSTRACT

Large Language Models (LLMs) exhibit impressive capabilities but often hallucinate, confidently providing incorrect answers instead of admitting ignorance. Prior work has shown that models encode linear representations of their own knowledge and that activation steering can reduce hallucinations. These approaches, however, require real-time monitoring and intervention during inference. We introduce **C**ontrastive **A**ctivation **S**teering for **A**mortized **L**earning (CASAL), an efficient algorithm that connects interpretability with amortized optimization. CASAL directly bakes the benefits of activation steering into model's weights during training. Once trained, LLMs answer questions they know while abstaining from answering those they do not. CASAL's light-weight design requires training only a sub-module of a single transformer layer and yet reduces hallucination by $\sim 30\%$ - $40\%$ across multiple short-form QA benchmarks. CASAL is $\sim 30$x more compute-efficient and $\sim 20$x more data-efficient than strong LoRA-based baselines such as SFT, DPO and GRPO, boosting its practical applicability in data scarce domains. Importantly, CASAL also generalizes effectively to out-of-distribution (OOD) domains. We showcase CASAL's flexibility in mitigating hallucinations in both text-only and vision-language models. To our knowledge, CASAL is the first steering-based training method that has been shown to be effective for both dense and Mixture-of-Experts (MoE) architectures. CASAL represents a significant step forward for applying interpretability-inspired method for scalable and broad deployment in production systems [1].

## 1 INTRODUCTION

Large Language Models (LLMs) have demonstrated near-human or even superhuman intellectual capabilities (Brown et al., 2020; Ouyang et al., 2022; OpenAI et al., 2024). Yet despite these successes, they sometimes fail in striking ways. A central failure mode is hallucination: the tendency to confidently generate false or unsupported information. Hallucinations undermine trust and restrict the safe deployment of LLMs in real-world settings where factual reliability is critical (Rawte et al., 2023; Gekhman et al., 2024a; Shen et al., 2025a).

Recent interpretability studies—using sparse autoencoder (SAE) features (Templeton et al., 2024) or residual stream activations (Rimsky et al., 2024; Turner et al., 2024)—have revealed that LLMs encode a form of self-knowledge. Specifically, the activations associated with known versus unknown knowledge can be separated along linear directions (Ji et al., 2025; Ferrando et al., 2025). Moreover, steering these representations reduces overconfidence and enables models to acknowledge uncertainty. However, prior work primarily focuses on inference-time interventions , leaving a significant gap in their practicality as part of scalable alignment pipelines.

If LLMs' internal states already reflect what is known versus unknown, why do they still produce confident but false answers? We hypothesize that a key cause lies in the training and evaluation

---

[1]https://github.com/facebookresearch/CASAL

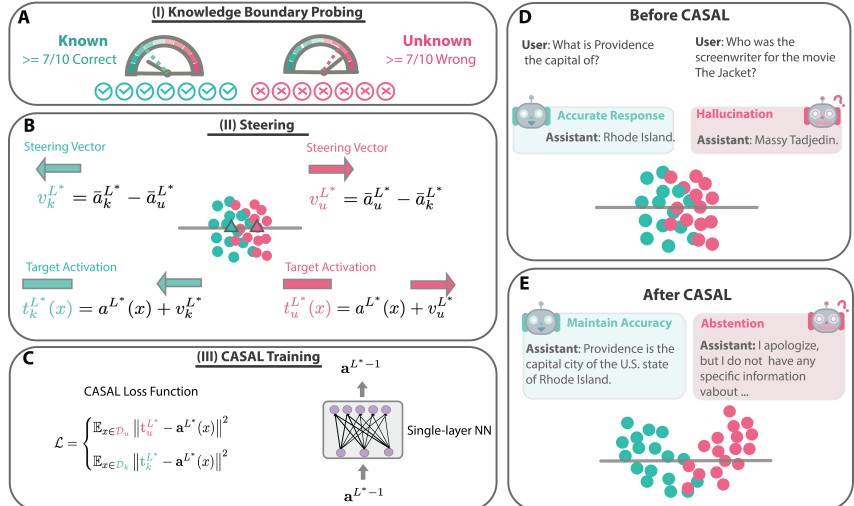

Figure 1: **Overview of the CASAL algorithm.** (A) **Knowledge Probing**: CASAL starts by probing the model to figure out what it knows vs doesn't know. Multiple responses per query are sampled to classify queries as known ($\mathcal{D}_k$) or unknown ($\mathcal{D}_u$). (B) **Steering**: Difference in means are computed to construct steering vectors ($\mathbf{v}_u^{L^*}$ and $\mathbf{v}_k^{L^*}$). Target activations ($\mathbf{t}_u^{L^*}$ and $\mathbf{t}_k^{L^*}$) are obtained by adding these steering vectors to the residual stream activation. **Pre-CASAL Behavior**: Prior to training, the model often hallucinates and produces incorrect answers for unknown queries. (C) **CASAL Training**: CASAL training is essentially "amortized activation steering", where instead of repeatedly steering activations online, we train a small subnetwork (a single layer NN) to approximate the steering solution offline. (D) **Post-CASAL Activations and Behavior**: After training, the model learns a sharper representation with a clearer knowledge boundary. It maintains correct answers on known queries while abstaining from answering unknown ones.

paradigm of LLMs (Li et al., 2025a; Kalai et al., 2025). During pretraining, the language modeling objective rewards predicting the next token given the training corpus distribution, incentivizing plausible continuations even under uncertainty rather than expressions of ignorance. Post-training further amplifies this tendency: the training and evaluation framework optimizes models to be good test-takers, rewarding guessing over acknowledging uncertainty (Gekhman et al., 2024b).

In this work, we propose an alternative training objective—one that leverages the model's own internal representations to align behavior with knowledge boundaries. Our core hypothesis is that if models are trained to **directly utilize their own representations** of known and unknown, their generations will better reflect what they truly "know". Concretely, we replace the standard cross-entropy loss with a local representation loss applied to residual stream activations. Whereas cross-entropy loss provides a learning signal from *external* supervision (the training corpus), representation loss provides a learning signal from *within*: model's own hidden representation.

Importantly, CASAL is among the first approaches to *rely solely on a representation-level objective* for training LLMs. Prior studies such as RepE (Zou et al., 2025), ReFAT (Yu et al., 2025), and others (Yu et al., 2024a; Casademunt et al., 2025; Chen et al., 2025b; Yousefpour et al., 2025) have explored representation-level fine-tuning, but all employed representation losses as auxiliary signals alongside standard cross entropy loss. By contrast, **CASAL treats representation loss as the only and the primary optimization objective**, directly teaching the model to utilize its hidden representation.

Our approach connects insights from two fields: interpretability and amortized optimization. Amortized optimization (Kingma and Welling, 2013; Rezende et al., 2014; Gershman and Goodman, 2014) is a paradigm where costly repeated optimizations are replaced by training a parametric function that approximates the solution. CASAL instantiates this idea by incorporating activation steering into training: **it "amortize" the activation steering process by training a lightweight subnetwork** that learns to approximate the steering solution, embedding the knowledge boundary directly into the model's weights.

---

**Algorithm 1** CASAL: Contrastive Activation Steering for Amortized Learning

---

**Require:** Dataset $\mathcal{D}$; frozen model $M_{\text{original}}$ with $l$ layers; target layer $L^*$; steering strength $\alpha$; training epochs $E$

    **STEP 1: Knowledge boundary probing**   **known** / **unknown**

1  Set $k = 10$, threshold $\tau = 7$
2  **for** $x \in \mathcal{D}$ **do**
3     Sample $k$ responses $\{y^{(i)}(x)\}$; $s(x) = \sum_i \mathbf{1}[y^{(i)}(x) \text{ correct}]$
4     **if** $s(x) \geq \tau$ **then** $\mathcal{D}_{\text{k}} \leftarrow \mathcal{D}_{\text{k}} \cup \{x\}$              ▷ "known" set $\mathcal{D}_{\text{k}}$
5     **else if** $k - s(x) \geq \tau$ **then** $\mathcal{D}_{\text{u}} \leftarrow \mathcal{D}_{\text{u}} \cup \{x\}$      ▷ "unknown" set $\mathcal{D}_{\text{u}}$
6     **end if**
7  **end for**

    **STEP 2: Steering**
    *Note:* $\mathbf{a}^{L^*}(x)$ denotes residual activations at layer $L^*$ for input $x$
8  $\bar{\mathbf{a}}_{\text{u}}^{L^*} = \frac{1}{|\mathcal{D}_{\text{u}}|} \sum_{x \in \mathcal{D}_{\text{u}}} \mathbf{a}^{L^*}(x), \quad \bar{\mathbf{a}}_{\text{k}}^{L^*} = \frac{1}{|\mathcal{D}_{\text{k}}|} \sum_{x \in \mathcal{D}_{\text{k}}} \mathbf{a}^{L^*}(x)$     ▷ mean activations
9  $\mathbf{v}_{\text{u}}^{L^*} = \bar{\mathbf{a}}_{\text{u}}^{L^*} - \bar{\mathbf{a}}_{\text{k}}^{L^*}, \quad \mathbf{v}_{\text{k}}^{L^*} = \bar{\mathbf{a}}_{\text{k}}^{L^*} - \bar{\mathbf{a}}_{\text{u}}^{L^*}$     ▷ steering vectors
10  $\mathbf{t}_{\text{u}}^{L^*}(x) = \mathbf{a}^{L^*}(x) + \alpha \cdot \mathbf{v}_{\text{u}}^{L^*}$ for $x \in \mathcal{D}_{\text{u}}$     ▷ "abstain when you don't know"
11  $\mathbf{t}_{\text{k}}^{L^*}(x) = \mathbf{a}^{L^*}(x) + \alpha \cdot \mathbf{v}_{\text{k}}^{L^*}$ for $x \in \mathcal{D}_{\text{k}}$     ▷ "answer when you know"

    **STEP 3: CASAL training**
12  Initialize one-layer network $M_{\text{train}}$ with weight $W_{\text{original}}^{L^*}$     ▷ one-layer fine-tuning
13  **for** $e = 1 \ldots E$ **do**
14     $\mathcal{L}_{\text{u}} = \mathbb{E}_{x \in \mathcal{D}_{\text{u}}} \|\mathbf{t}_{\text{u}}^{L^*}(x) - \mathbf{a}^{L^*}(x)\|^2$     ▷ "unknown" loss
15     $\mathcal{L}_{\text{k}} = \mathbb{E}_{x \in \mathcal{D}_{\text{k}}} \|\mathbf{t}_{\text{k}}^{L^*}(x) - \mathbf{a}^{L^*}(x)\|^2$     ▷ "known" loss
16     $\mathcal{L} \leftarrow \mathcal{L}_{\text{u}} + \mathcal{L}_{\text{k}}$; update $M_{\text{train}}$ weights by $\nabla \mathcal{L}$
17  **end for**
18  $W_{\text{CASAL}}^{L^*} \leftarrow$ trained weights from $M_{\text{train}}$     ▷ extract trained weights
19  $M_{\text{CASAL}} \leftarrow M_{\text{original}}$ with $W_{\text{original}}^{L^*}$ replaced by $W_{\text{CASAL}}^{L^*}$ at layer $L^*$     ▷ create output model
**Ensure:** Trained model $M_{\text{CASAL}}$ with updated weights at layer $L^*$

---

We highlight our main contributions as:

- **Effective Algorithm**: Introducing a training method inspired by interpretability findings and amortized optimization. CASAL enables models to admit ignorance for unknown questions, reducing hallucination rates by $\sim 30\%$ - $40\%$ across multiple short-form QA benchmarks.

- **Efficiency Gains**: CASAL's objective function enables local and lightweight parameter updates, delivering $\sim$ **30x higher compute efficiency (FLOPs per token)** and requires $\sim$**20x less training data** (with as little as $\sim 640$ training data) to achieve the same level of performance compared to LoRA-based SFT and DPO.

- **Robust Generalization**: The trained model retains its general capabilities while avoiding excessive refusals. At the same time, it successfully generalizes refusal behavior to unknown queries sampled from out-of-distribution (OOD) data.

- **Versatility**: CASAL training is modality-agnostic, effectively mitigating hallucination in both text-only and **multimodal models**.

- **Broad Applicability**: We present *the first* ever steering-based training framework with *general* applicability to both **dense and Mixture-of-Experts (MoE) models.**

## 2   CASAL

We now introduce our method, CASAL, which integrates insights from interpretability and amortized optimization to build a lightweight, efficient training framework. The full pipeline is shown in Figure 1, summarized in Algorithm 1. At a high level, CASAL can be understood as an instance

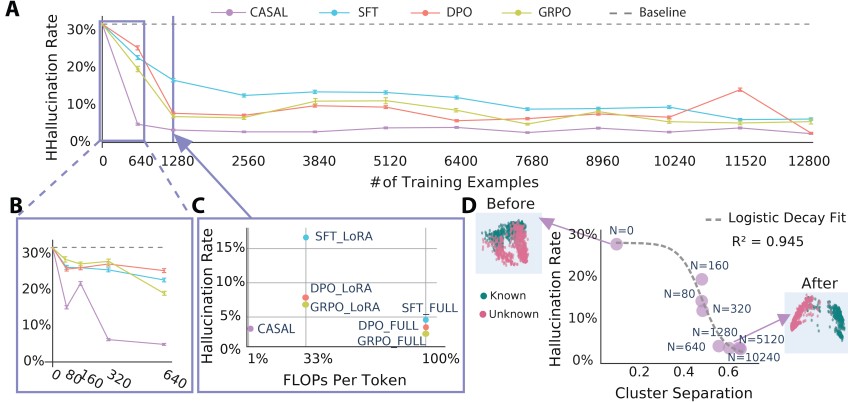

Figure 2: **CASAL is both sample efficient and compute efficient.** (A–B) CASAL achieves strong hallucination reduction with orders-of-magnitude fewer training examples comparing to LoRA-based fine-tuning with SFT, DPO and GRPO. (C) CASAL is over $30\times$ more compute-efficient than PEFT baselines such as LoRA. (D) Hallucination reduction after CASAL training correlates with improved cluster separation between known and unknown queries, measured by silhouette score.

of *amortized optimization*: instead of repeatedly solving the steering problem at inference time, we train a parametric subnetwork to approximate this solution once, thereby "amortizing" the resource use of activation steering across all future queries. This perspective motivates the name: **C**ontrastive **A**ctivation **S**teering for **A**mortized **L**earning (CASAL). CASAL proceeds in three stages:

## 2.1 STEP 1: KNOWLEDGE BOUNDARY PROBING

CASAL begins by probing the model to delineate its knowledge boundary. For each input $x \in \mathcal{D}$, we sample $k = 10$ completions and compare them to ground-truth answers. For each question, if at least 7 generations are correct, $x$ is labeled as known; if less than 3 are incorrect, it is labeled as unknown. This produces two subsets: $\mathcal{D}_k$ and $\mathcal{D}_u$. We systematically evaluated different threshold values and found that hallucination reduction performance remains robust across this range (Appendix I). We adopt a relatively strict threshold of $\tau = 7$ to ensure high-confidence separation: the model abstains only on knowledge it does not possess, and responds only when it demonstrates consistent correctness. This choice reduces ambiguous cases near the decision boundary. [2] We evaluate CASAL on three datasets: TriviaQA (Joshi et al., 2017b), PopQA (Mallen et al., 2023b), and EntityQA (Ferrando et al., 2025). Dataset details provided in Appendix G.1.

## 2.2 STEP 2: STEERING

Next, CASAL constructs contrastive steering vectors to obtain better knowledge boundaries (Rimsky et al., 2024; Turner et al., 2024; Arditi et al., 2024). For each query $x$, we extract residual stream activations $\mathbf{a}^{L^*}(x)$ at a designated target layer $L^*$ from *the last token position of the question*[3]. We then compute mean activations for known and unknown subsets ($\bar{a}_k^{L^*}$ and $\bar{a}_u^{L^*}$) and construct steering vectors by taking difference in means, resulting in two vectors: $\mathbf{v}_u^{L^*}$ for abstaining when the model lacks knowledge, and $\mathbf{v}_k^{L^*}$ for reinforcing correct answering when the model does know. The steering vectors are then added to the residual stream activations, yielding target activations $\mathbf{t}_u^{L^*}$ and $\mathbf{t}_k^{L^*}$. These target activations are cached and subsequently used to compute the representation loss in STEP 3. Further details for steering and target layer selection procedures are included in Appendix C.

---

[2]Consistent with previous literature (Ferrando et al., 2025; Grattafiori et al., 2024), the knowledge probing step creates the known versus unknown labels subsequently used for steering our training baseline methods such as SFT and DPO, and therefore does not introduce additional computational cost specific to CASAL.

[3]By extracting activations from the last token position of the *question*, the steering vectors reflect properties of the question itself (whether it is known or unknown to the model) rather than features of the *answer* (whether the answer is correct or incorrect).

## 2.3 STEP 3: CASAL TRAINING

Finally, CASAL trains a lightweight **one-layer network** $M_{\text{train}}$, initialized with the weight $W_{\text{original}}^{L^*}$ from layer $L^*$ of the original model. Using a mean squared error objective, CASAL minimizes the distance between current activation $\mathbf{a}^{L^*}(x)$ and its corresponding target activation ($\mathbf{t}_k^{L^*}$ or $\mathbf{t}_u^{L^*}$). After training, the learned weights $W_{\text{CASAL}}^{L^*}$ are extracted from $M_{\text{train}}$ and substituted back into layer $L^*$ of the original model, producing the final model $M_{\text{CASAL}}$. This process embeds the knowledge boundary directly into the model weights, eliminating the need for repeated steering at inference. **Importantly, this representation loss is the sole training objective**, not used as auxiliary loss with standard cross-entropy. Because this loss is **local to layer** $L^*$ (derived directly from residual activations at that layer), we only need to train one single layer. This contrasts with cross-entropy loss, which requires a forward pass through all layers to compute output probabilities. Even when updating only a single target layer with cross-entropy loss, the entire model (with other layers frozen) must be deployed during the forward pass, adding much more computational cost compared to training just the one-layer network $M_{\text{train}}$. We conducted systematic ablation studies (Appendix K) to examine different fine-tuning strategies. Our results demonstrate that fine-tuning different submodules of the MLP layer yields no statistically significant performance differences. Further details for the training process and hyperparameter research are included in Appendix D and L.

## 3 CASAL IS EFFECTIVE AND EFFICIENT

We evaluate CASAL against strong baselines including Supervised Fine-Tuning (SFT), Direct Preference Optimization (DPO) and Group Relative Policy Optimization (GRPO), which represent the predominant fine-tuning approaches deployed in production systems today (hyperparameters search and other training details are provided in Appendix M). By demonstrating CASAL's superiority over these widely-adopted techniques, we establish its practical applicability for real-world deployment beyond toy settings.

### 3.1 SAMPLE EFFICIENCY

We quantify hallucination reduction performance primarily using the *hallucination rate*, which captures the fraction of unknown queries incorrectly attempted by the model. Figure 2 summarizes our key findings, with additional details on the hallucination rate metric provided in Appendix H.2. CASAL achieves substantially lower hallucination rates across a wide range of training set sizes. When trained on just 640 examples, CASAL already matches or surpasses the performance of SFT, DPO and GRPO trained on 12,800 examples (Figure 2A–B). This translates into more than **20×** **higher data efficiency**, demonstrating that CASAL is especially practical in data-scarce settings.

### 3.2 COMPUTE EFFICIENCY

Beyond sample efficiency, CASAL is also highly compute efficient. By updating only a lightweight sub-module within a single transformer layer, CASAL is substantially more compute efficient than full fine-tuning or even LoRA-based parameter-efficient fine-tuning (PEFT). As shown in Figure 2C, CASAL achieves lower hallucination rates while requiring over **30×** **fewer FLOPs per token** than LoRA during training, underscoring its practicality for large-scale deployments. This efficiency stems from two key properties of CASAL's loss function:

**Efficiency across model depth.** Because CASAL's loss is local to layer $L^*$, both forward and backward passes operate exclusively within the single-layer network $M_{\text{train}}$. In contrast, methods using cross-entropy loss, even when updating only a single layer with other layers frozen, must perform a forward pass through all layers end-to-end to compute output probabilities and backpropagate gradients from the output back to the target layer. For example, when fine-tuning layer 16 of a 32-layer model, cross-entropy-based methods require computations through 32 layers in the forward pass and through 16 layers in the backward pass, while CASAL operates only on the target layer itself. This advantage scales with model depth: the deeper the model, the greater CASAL's computational savings.

**Efficiency across generation length.** CASAL computes loss at a single position—the last token of the question. In contrast, SFT averages cross-entropy loss over *all tokens* in the generated answer,

while DPO computes log-probabilities over *all tokens* in both chosen and rejected responses. The computational cost thus scales with answer length for these methods, whereas CASAL's cost remains constant regardless of generation length. Longer answers make CASAL increasingly cost-effective comparing to standard baselines.[4] Details of FLOPs calculations are included in Appendix N.

### 3.3 LEARNING BETTER KNOWLEDGE BOUNDARIES

By training with a local representation loss, CASAL encourages clearer separation between activations corresponding to known and unknown queries. We compute Silhouette score as a measure of cluster separation. As shown in Figure 2D, Silhouette scores ( H.4). increase as training progresses, and this separation is correlated with the reduction in hallucination rate. The strong correspondence (logistic fit, $R^2 = 0.945$) between representational separation and behavioral outcomes indicates that CASAL's effectiveness arises from more faithfully encoding and utilizing knowledge boundaries. Consistent with our hypothesis, CASAL demonstrates the best cluster separation and the clearest boundary between known and unknown queries compared to the other methods ( Figure 22, Appendix Q). This validates that by directly training a local representation loss, CASAL effectively encourages a distinct separation between these activation states.

| Methods | Refusal Rate (↓) | | | Accuracy (↑) | | |
|---|---|---|---|---|---|---|
| | **PopQA** | **TriviaQA** | **EntityQA** | **PopQA** | **TriviaQA** | **EntityQA** |
| Baseline | **18.19%±3.01** | 7.93%±1.14 | 8.94%±2.18 | **91.08%±2.23** | **95.82%±2.24** | 88.59%±1.46 |
| SFT | 20.32%±1.09 | 10.01%±1.16 | 11.08%±1.24 | 82.89%±1.33 | 92.45%±1.29 | 85.75%±1.18 |
| DPO | 21.79%±1.11 | 14.37%±2.06 | 17.66%±2.14 | 90.25%±1.06 | 95.30%±0.96 | 89.84%±1.16 |
| GRPO | 17.48%±4.46 | 17.77%±3.82 | 16.67%±4.42 | 85.78%±4.36 | 91.67%±2.76 | 85.48%±3.52 |
| CASAL | 19.89%±1.15 | **7.29%±1.34** | **6.84%±1.23** | 85.11%±1.88 | 95.34%±2.25 | **89.90%±0.99** |

Table 1: CASAL does not introduce over-refusal nor degrade performance for known queries. Refusal rate and accuracy across three different QA datasets are measured.

| Methods | Accuracy (↑) | | | Win rate (↑) |
|---|---|---|---|---|
| | **MMLU (General)** | **GSM8K (Math)** | **GPQA (Reasoning)** | **MT Bench (Coherence)** |
| Baseline | 68.01 ± 0.34 | 77.48 ± 1.15 | **33.31 ± 0.34** | 7.38 ± 0.06 |
| SFT | 67.90 ± 0.23 | 75.66 ± 1.18 | 32.82 ± 0.34 | 7.44 ± 0.13 |
| DPO | 68.03 ± 0.26 | **78.16 ± 1.14** | 31.43 ± 0.37 | 7.39 ± 0.15 |
| GRPO | 67.73 ± 0.38 | 76.66 ± 1.21 | 31.92 ± 0.22 | 7.44 ± 0.11 |
| CASAL | **68.04 ± 0.44** | 77.02 ± 1.16 | 33.18 ± 0.34 | **7.57± 0.08** |

Table 2: CASAL preserves general capability. Performances (higher is better) on general capability, math, reasoning and context-aware conversational ability in multi-turn dialogues are measured.

### 4 CASAL PRESERVES MODEL CAPABILITY

An important requirement for any practically useful hallucination-reduction method is that it should not degrade a model's general capabilities nor induce excessive refusals on queries the model can correctly answer. We therefore evaluate CASAL across both refusal behavior and broad capability benchmarks. Table 1 reports refusal rates on three QA datasets. CASAL achieves the lowest refusal rates on TriviaQA (7.29%) and EntityQA (6.84%), while maintaining a competitive rate on PopQA (19.89%). These results demonstrate that CASAL reduces hallucination on unknown queries without over-penalizing the model into unnecessary refusals for known ones. We also evaluate against

---

[4]The FLOPs comparison reported in this work is measured *per token*. This makes our estimate of CASAL's computational advantage (30× fewer FLOPs than LoRA) conservative. For tasks requiring longer generations, CASAL's efficiency gains over SFT, DPO and GRPO would be substantially greater.

| Dataset | Methods | Hallucination Rate (Unknown) ($\downarrow$) | | Refusal Rate (Known) ($\downarrow$) | | Accuracy (Known) ($\uparrow$) | |
|---|---|---|---|---|---|---|---|
| | | **Train** | **Test** | **Train** | **Test** | **Train** | **Test** |
| TriviaQA | | **Wiki** | **Web** | **Wiki** | **Web** | **Wiki** | **Web** |
| | Baseline | 48.20%±1.34 | 50.74%±1.12 | 9.06%±0.93 | 7.93%±2.02 | 94.22%±1.44 | 95.82%±1.22 |
| | SFT | 24.44%±1.65 | 35.44%±1.64 | 14.77%±1.52 | 15.10%±1.77 | 91.23%±0.54 | 90.26%±1.15 |
| | DPO | 23.28%±1.02 | 33.77%±0.99 | 13.62%±1.08 | 16.33%±1.19 | 90.23%±1.09 | 88.13%±1.30 |
| | GRPO | 22.33%±1.34 | 33.12%±1.88 | 15.99%±1.30 | 18.10%±1.09 | 88.83%±0.96 | 87.27%±1.03 |
| | CASAL | **20.47%±1.11** | **32.42%±1.29** | **8.28%±1.16** | 11.69%±2.22 | 92.03%±0.82 | 90.08%±1.33 |
| PopQA | | **Group 1** | **Group 2** | **Group 1** | **Group 2** | **Group 1** | **Group 2** |
| | Baseline | 74.87%±2.92 | 74.35%±1.56 | 18.95%±1.39 | 18.19%±1.46 | 90.84%±1.51 | 91.08%±0.92 |
| | SFT | 22.77%±1.08 | 24.02%±1.11 | 14.04%±1.16 | 20.88%±1.09 | 85.01%±1.06 | 85.74%±1.60 |
| | DPO | 21.08%±1.33 | 24.88%±0.99 | 14.99%±1.62 | 19.19%±1.55 | 84.66%±0.90 | 84.01%±1.11 |
| | GRPO | 20.19%±1.05 | 24.22%±1.32 | 18.07%±1.02 | 21.10%±1.33 | 80.13%±0.31 | 80.98%±1.06 |
| | CASAL | **22.48%±1.45** | **23.42%±1.94** | **13.97%±1.78** | 19.10%±1.30 | 85.23%±0.86 | 84.27%±1.99 |

Table 3: CASAL learns a generalizable notion of known vs. unknown, and can transfer between data sources within TriviaQA and generalize across groups within PopQA.

Contrastive Activation Addition (CAA), a popular inference-time steering method (Rimsky et al., 2024). As summarized in Section E, while CASAL achieves comparable hallucination rates to CAA on unknown queries, it maintains performance on known queries, whereas CAA degrades accuracy for questions the model could previously answer correctly. This finding aligns with previous work (Durmus et al., 2024; Chen et al., 2025b) showing that inference-time steering can introduce undesirable side effects.

We further assess models' general capability, including MMLU (Hendrycks et al., 2021) for general knowledge, GSM8K (Cobbe et al., 2021) for math reasoning, GPQA(Rein et al., 2023) for scientific reasoning, and MT-Bench (Zheng et al., 2023) for coherence in multi-turn conversations. As shown in Table 2, CASAL performs on par with strong baselines across all metrics. Beyond these quantitative measures, we provide raw model outputs in Appendix F to allow readers to assess the natural flow and coherence of generated responses after CASAL training. These results demonstrate that CASAL reduces hallucinations on unknown queries while avoiding over-refusal on known queries, all without sacrificing general capability—a balance critical for practical deployment.

## 5 CASAL IS OOD GENERALIZABLE

Does CASAL capture a generalizable notion of what the model knows versus does not know beyond its training distribution? We test its ability to generalize across both in-distribution and out-of-distribution (OOD) settings. We first evaluate whether CASAL's learned knowledge boundary transfers across different groups within the same dataset. As shown in Table 3, CASAL trained on Wikipedia-style data generalizes effectively to web data, reducing hallucination rate from 50.7% to 32.4% while maintaining high accuracy on known queries (92.0% vs. 95.8%). A similar trend is observed on PopQA (Table 3), where CASAL substantially reduces hallucinations in both Group 1 and Group 2, lowering test hallucination rates from 74.4% to 23.4%. These results indicate that CASAL does not simply memorize steering directions but learns a transferable notion of known versus unknown knowledge that holds across diverse data groups.

We next evaluate a stronger OOD setting: training CASAL on one dataset and testing it on a completely different one. Specifically, CASAL is trained on TriviaQA and evaluated on EntityQA (Table 4). Remarkably, hallucination rate on the unseen EntityQA dataset drops from 50.7% to 11.7%, while accuracy on known queries remains above 95%. This demonstrates that CASAL's learned representations extend beyond the training domain, capturing knowledge boundaries that remain robust even under OOD transfer. Together, these results establish that CASAL generalizes well both across sub-groups within a dataset and across entirely distinct datasets. This robustness highlights that CASAL is not merely overfitting to a narrow training distribution but instead induces a broadly applicable mechanism for distinguishing known from unknown queries.

| Methods | Hallucination Rate (↓) | | Refusal Rate (↓) | | Accuracy (↑) | |
|---|---|---|---|---|---|---|
| | **Train** | **Test** | **Train** | **Test** | **Train** | **Test** |
| | **TriviaQA** | **EntityQA** | **TriviaQA** | **EntityQA** | **TriviaQA** | **EntityQA** |
| Baseline | 48.2%±1.33 | 50.74%±0.92 | **9.06%±1.22** | **12.89%±1.49** | **94.22%±0.93** | **95.82%±2.32** |
| SFT | 30.63%±1.53 | 23.13%±1.66 | 21.16%±1.34 | 29.84%±1.40 | 88.80%±1.33 | 80.77%±1.05 |
| DPO | 20.63%±2.21 | 18.30%±1.45 | 13.89%±1.22 | 22.02%±1.29 | 92.91%±1.06 | 87.41%±1.22 |
| GRPO | **14.00%±0.55** | 12.83%±4.18 | 21.43%±4.10 | 17.25%±5.56 | 85.69%±8.75 | 85.24%±4.32 |
| CASAL | 18.23%±1.03 | **11.72%±1.66** | 9.29%±1.48 | 13.82%±1.55 | 93.36%±1.47 | 95.77%±1.64 |

Table 4: CASAL supports OOD generalization across different datasets. The model is trained on the TriviaQA dataset and tested on EntityQA as an out-of-distribution setting.

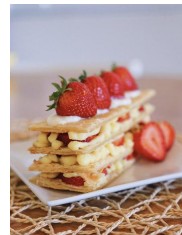

*What is this dish known as in France?*

| Methods | WorldCuisines Dataset | | |
|---|---|---|---|
| | Unknown | Known | |
| | **Hallucination Rate (↓)** | **Refusal Rate (↓)** | **Accuracy (↑)** |
| Baseline | 72.35%±1.77 | **13.91%±1.37** | 76.72%±1.67 |
| SFT | 35.05%±2.87 | 24.33%±2.76 | 87.42%±1.66 |
| DPO | 36.44%±3.77 | 24.02%±2.11 | 86.66%±1.74 |
| GRPO | 35.19%±2.99 | 28.73%±2.73 | 80.18%±1.64 |
| CASAL | **33.34%±3.13** | 25.44%±2.91 | **90.36%±1.96** |

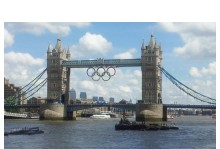

*Where is this place?*

| Methods | Landmark Dataset | | |
|---|---|---|---|
| | Unknown | Known | |
| | **Hallucination Rate (↓)** | **Refusal Rate (↓)** | **Accuracy (↑)** |
| Baseline | 75.78%±1.69 | 3.59%±0.73 | 90.80%±1.55 |
| SFT | 39.05%±4.07 | 2.64%±3.01 | 92.77%±1.42 |
| DPO | 35.99%±1.03 | 6.06%±1.44 | 94.11%±1.01 |
| GRPO | 35.44%±1.81 | 8.19%±1.09 | 90.75%±1.31 |
| CASAL | **31.25%±8.32** | **3.12%±3.01** | **99%±0.03** |

Table 5: **CASAL is modality agnostic.** It reduces hallucination in vision-language model on WorldCuisines-VQA (top) and Landmark-VQA (bottom). Example question-image pairs from the two datasets are shown on the left.

## 6 CASAL IS MODALITY AND ARCHITECTURE AGNOSTIC

### 6.1 CASAL REDUCES HALLUCINATION IN VISION-LANGUAGE MODELS

We apply CASAL to a vision-language model: Qwen2.5-VL-7B-Instruct (Qwen et al., 2024) and perform training on the WorldCuisines-VQA (Winata et al., 2024) dataset. Finally, we evaluate whether CASAL generalizes beyond standard dense transformer architectures and text-only settings. CASAL reduces hallucination rate (Table 5) by 38.74%. Importantly, accuracy on known queries is preserved. This confirms that CASAL's mechanism for sharpening knowledge boundaries is not tied to language-only models but extends naturally to multimodal models. Further details for training vision-language models are provided in Appendix O.

### 6.2 CASAL REDUCES HALLUCINATION IN MIXTURE-OF-EXPERTS MODELS

MoE models pose a unique challenge since knowledge and uncertainty may be distributed across different experts. We first ask "how are unknown versus known queries represented across experts?" Are certain experts specialized in representing known and others specialized in unknown? Or are they co-represented in the same experts? We started our investigation by visualizing the activations in different experts in the OLMoE model (Muennighoff et al., 2025). As illustrated in Figure 18A,

activations for known and unknown queries are mostly co-represented in the same experts. Similar to dense model training, CASAL applies a local representation loss on the residual stream activations with converging signal across all experts (Figure 18B). After training, residual stream activations show a much clearer boundary between known and unknown queries (Figure 18C), which translates into significant improvements in hallucination rates. Hallucination rate for unknown queries drops by 42.9%, while accuracy on known queries remains unchanged (Figure 18D). Further details regarding the CASAL training for MoE models can be found in Appendix P. These results demonstrate that CASAL effectively extends to MoE architectures without sacrificing accuracy. Together, these results establish that CASAL is both *architecture-agnostic* and *modality-agnostic*. Whether applied to dense or MoE transformers, or to text-only versus vision-language models, CASAL consistently reduces hallucination rates while maintaining high accuracy and balanced refusal behavior. This broad applicability highlights CASAL's potential as a scalable, general-purpose alignment technique.

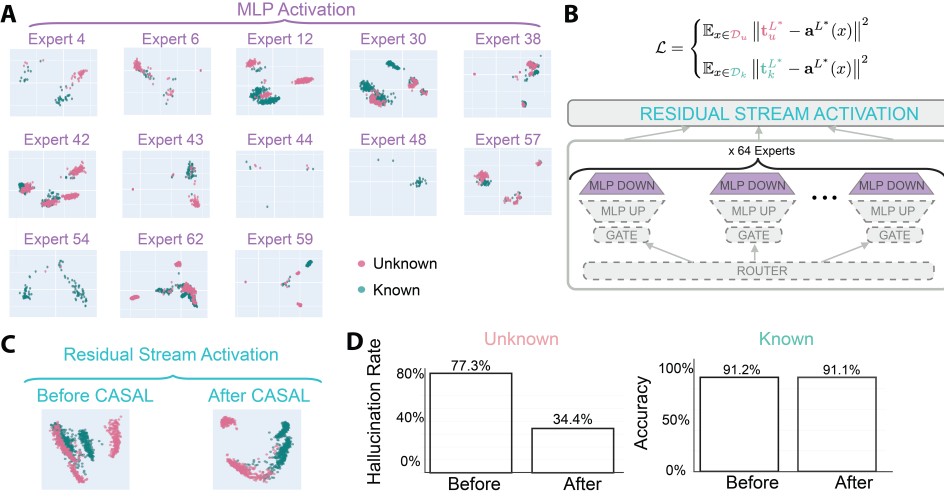

Figure 3: **CASAL is architecture-agnostic.** It effectively reduces hallucination for OLMoE. (A) Visualization of MLP activations from different experts in a MoE model before CASAL training. (B) CASAL applies a local representation loss on residual stream activations. During training, weights are updated on only a lightweight sub-module across experts. (C) Residual stream activations before and after CASAL training. (D) CASAL reduces hallucination rate on unknown queries while maintaining low refusal score and high accuracy for known queries.

# 7 RELATED WORK

## 7.1 HALLUCINATION MITIGATION

**Inference-time Intervention.** Steering-based approaches (Rimsky et al., 2024; Turner et al., 2024) for hallucination reduction typically apply interventions during inference (Ferrando et al., 2025; Ji et al., 2025; Li et al., 2024; Park et al., 2025). While effective, this requires solving a local optimization problem for every input (e.g., shifting activations along a direction at every forward pass), introducing extra computational overhead during deployment to monitor and intervene. In contrast, CASAL eliminates the need for per-instance intervention by directly baking the knowledge boundaries into model parameters, enabling scalable deployment in production.

**In-weight Learning.** A complementary body of work modifies model parameters to encourage calibrated abstention and reduce hallucination. Early approaches train models to abstain from uncertain predictions via probabilistic calibration. Others focus on eliciting explicit confidence estimates in conversational models (Chen et al., 2024; Mielke et al., 2022). Concurrent work Chen et al. (2025b) proposes persona vector extraction, where finetuning steers models away from undesired persona directions. CASAL differs in two key ways: (i) rather than steering *away* from undesirable traits, we explicitly steer *towards* desirable representations; and (ii) CASAL presents an

efficient training framework, yielding ~$30\times$ higher compute efficiency than SOTA parameter-efficient finetuning methods such as LoRA.

## 7.2 AMORTIZED OPTIMIZATION, ACTIVATION STEERING AND REPRESENTATION LEARNING

**Amortized Optimization.** Amortized optimization (Kingma and Welling, 2013; Rezende et al., 2014; Gershman and Goodman, 2014) is a widely used paradigm in which expensive, repeated optimization is replaced by training a parametric function that approximates the solution. Despite its influence in areas such as variational inference, sparse coding, gradient-based meta-learning and reinforcement learning (Amos, 2025; Chen et al., 2021), this perspective has been explored less in the context of interpretability or alignment (Paulus et al., 2025). CASAL can be viewed as *amortized activation steering*, where the resource intensive process of online steering is distilled into a lightweight subnetwork trained offline and reused at inference.

**Activation Steering.** A line of work has focused on inference-time interventions, where steering vectors are applied dynamically to control model behavior without modifying weights (Ji et al., 2025; Li et al., 2024). Within this paradigm, a common approach to derive steering vectors is to construct sample pairs differing along a target concept and compute their difference-in-means (Arditi et al., 2024). Alternative methods further fine-tune the steering vectors to enable more effective behavior control with less side effect (Cao et al., 2024; Stickland et al., 2024; Parekh et al., 2025). Another line of work leverages sparse autoencoders (SAEs) to uncover interpretable features in an unsupervised manner, which can then serve as handles for steering interventions (Ferrando et al., 2025).

**Representation Learning.** A parallel line of work (Tian et al., 2025; Yu et al., 2024a; Chen et al., 2025b; Casademunt et al., 2025) focuses on shaping internal representations during finetuning to suppress undesired behaviors. Early methods include representation fine-tuning (ReFT), which encourages task-specific interventions on hidden states (Wu et al., 2024), and representation engineering (RepE), which monitors and manipulates high-level cognitive phenomena in LLMs (Zou et al., 2025). Other techniques explicitly control harmful states: Zou et al. (2024) introduce circuit breakers to block dangerous representations, while Yu et al. (2025) perform directional ablation of refusal features to maintain robustness under adversarial attacks. Similarly, Yousefpour et al. (2025) propose representation bending to disrupt harmful latent features. For unlearning, Shen et al. (2025b) train models to redirect unlearning data into refusal regions. Compared to these efforts, CASAL provides *the first* general steering-based training framework that is broadly applicable to **both dense and sparse (MoE) architectures**.

## 8 CONCLUSION AND LIMITATIONS

In this work, we introduced CASAL, a lightweight, effective, and broadly applicable method for reducing hallucinations in large language models. By embedding knowledge boundaries directly into model weights, CASAL achieves substantial reductions in hallucination without degrading general capabilities, while being markedly more compute- and data-efficient than standard baselines. Beyond its empirical results, CASAL provides initial evidence a broader principle: insights from interpretability can be distilled into training objectives that scale.

While CASAL shows strong effectiveness and efficiency, several limitations remain. First, although CASAL generalizes across short-form QA datasets, modalities, and architectures, its effectiveness in reasoning models remains to be systematically tested. Second, our evaluation focuses specifically on hallucinations in short-form QA tasks. Exploring CASAL's effectiveness in reducing hallucinations during long-form generations (Obeso et al., 2025) represents an important direction for future research. Finally, one particularly exciting future direction is the integration of CASAL into LLM-based agentic systems. As LLMs move toward becoming tool-using agents integrated into everyday workflows, their reliability becomes critical—misplaced confidence can lead to cascading errors with tangible consequences. While modern agents increasingly leverage external tools to address factual uncertainty, effective tool orchestration fundamentally depends on the agent's ability to recognize the boundaries of its own knowledge. CASAL's mechanism for sharpening these knowledge boundaries could therefore serve as a component for more reliable agentic systems, enabling agents to make better decisions about when to respond directly versus when to invoke tools such as web search or specialized knowledge bases.

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

## 9 ACKNOWLEDGEMENT

We also thank Nathaniel Li, Irene Zhang, Julian Coda-Forno, Sriyash Poddar, Anselm Paulus, Sai Surya Duvvuri, Rachit Bansal, Devvrit Khatri, Ellie Pavlick and Jojo Yang for providing thoughtful feedback and insightful discussions on the manuscript.

APPENDIX

# Table of Contents

# A FURTHER DISCUSSION ON RELATED WORK

## A.1 KNOWLEDGE REPRESENTATION AND THE LINEAR REPRESENTATION HYPOTHESIS

Humans often display systematic overconfidence: their subjective confidence often exceeds objective accuracy (Pallier et al., 2002; Stankov and Crawford, 1996). Large language models (LLMs) exhibit a similar pattern: they are poorly calibrated on general knowledge tasks, frequently producing answers with misplaced confidence (Kadavath et al., 2022; Yin et al., 2023b; Yona et al., 2024; Zhang et al., 2025).

Recent interpretability studies (using sparse autoencoder (SAE) features (Ferrando et al., 2025) or residual stream activations (Ji et al., 2025)) suggest that transformer models encode many abstract concepts as linear directions in activation space (Nanda et al., 2023; Mikolov et al., 2013; Park et al., 2023; Arditi et al., 2024; Yang and Buzsáki, 2025). Behavioral traits such as truthfulness, sycophancy, refusal (Arditi et al., 2024), and reasoning strategies have shown to be linearly represented. Emerging evidence indicates that models may also possess intrinsic linear representations of knowledge boundary (Ferrando et al., 2025) and uncertainty (Ji et al., 2025) for their own knowledge limitation, which can be harnessed for calibrating overconfidence in LLMs.

# B FURTHER DISCUSSION ON AMORTIZED OPTIMIZATION

**Amortized Optimization Perspective.** Our approach combines insights from interpretability and amortized optimization (Kingma and Welling, 2013; Rezende et al., 2014; Gershman and Goodman, 2014). Formally, amortized optimization replaces repeated problem-specific optimizations

$$\theta^*(x) = \arg\min_{\theta} \ \mathcal{L}(f_\theta, x)$$

with the training of a parametric function $g_\phi(x)$ that directly predicts an approximate solution, i.e., $\theta^*(x) \approx g_\phi(x)$. This paradigm reduces per-instance optimization compute cost by learning a global set of parameters $\phi$ that amortize inference across the data distribution.

VAEs provide a canonical example: instead of optimizing a separate variational posterior $q(z|x)$ for every datapoint, the encoder $q_\phi(z|x)$ is trained to amortize inference. The optimization signal is the evidence lower bound (ELBO),

$$\mathcal{L}_{\text{ELBO}}(\theta, \phi) = \mathbb{E}_{q_\phi(z|x)}[\log p_\theta(x|z)] - \text{KL}(q_\phi(z|x) \,\|\, p(z)),$$

Amortization arises from the parameterization of inference with a shared encoder network $q_\phi(z|x)$, which maps each input x to distributional parameters in a single forward pass, replacing the need to optimize separate variational parameters for each datapoint.

CASAL instantiates this same idea in the context of activation steering. Instead of repeatedly solving for a steering direction $v^*(x)$ that separates known from unknown knowledge in residual activations $h(x)$, we train a lightweight subnetwork $s_\phi$ to approximate this solution:

$$v^*(x) \ \approx \ s_\phi(h(x)).$$

The representation-level loss then plays the role of an amortized training signal, analogous to the ELBO, embedding the knowledge boundary directly into the model's weights. This allows the model to align its outputs with its internal representations in a single forward pass, making steering efficient and scalable.

## C  STEERING

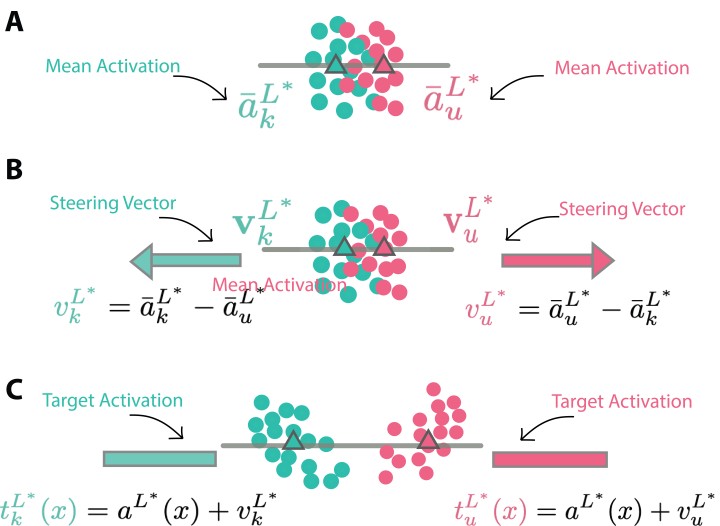

Figure 4: **Illustration of steering vector and target activation construction.** (A) Mean activations at the target layer $L^*$ are computed for known queries ($\bar{a}_k^{L^*}$) and unknown queries ($\bar{a}_u^{L^*}$). (B) Steering vectors are defined by the difference of these means: $v_k^{L^*} = \bar{a}_k^{L^*} - \bar{a}_u^{L^*}$ (pointing toward the known cluster) and $v_u^{L^*} = \bar{a}_u^{L^*} - \bar{a}_k^{L^*}$ (pointing toward the unknown cluster). (C) Target activations are generated by shifting the raw activations $a^{L^*}(x)$ along the corresponding steering vector: $t_k^{L^*}(x) = a^{L^*}(x) + v_k^{L^*}$ for known queries, and $t_u^{L^*}(x) = a^{L^*}(x) + v_u^{L^*}$ for unknown queries. These target activations serve as supervision signals during CASAL training.

### C.1  STEERING VECTOR CONSTRUCTION

**Known vs. Unknown Separation.** Queries are partitioned into $\mathcal{D}_k$ (known) and $\mathcal{D}_u$ (unknown) based on the model's consistency across multiple sampled answers. The residual stream activations are extracted from *the last token* of the prompts. Averaged activations over each set yield mean activations:

$$\bar{a}_k^{L^*} = \mathbb{E}_{x \in \mathcal{D}_k}[a^{L^*}(x)], \quad \bar{a}_u^{L^*} = \mathbb{E}_{x \in \mathcal{D}_u}[a^{L^*}(x)].$$

**Steering Vectors and Target Activations.** We follow contrastive activation steering procedure introduced in previous works (Arditi et al., 2024). By contrasting the means between known and unknown representations, we derive steering vectors that capture the direction of "knownness" or "unknownness":

$$v_u^{L^*} = \bar{a}_u^{L^*} - \bar{a}_k^{L^*}, \quad v_k^{L^*} = \bar{a}_k^{L^*} - \bar{a}_u^{L^*}.$$

Applying these shifts to an activation produces *target activations*:

$$t_u^{L^*}(x) = a^{L^*}(x) + v_u^{L^*}, \quad t_k^{L^*}(x) = a^{L^*}(x) + v_k^{L^*}.$$

Intuitively, $t_u^{L^*}(x)$ encourages the model to abstain when uncertain, while $t_k^{L^*}(x)$ reinforces confident answering when the knowledge is present.

### C.2  LAYER SELECTION

A crucial step in CASAL is selecting the optimal target layer $L^*$. To identify this layer, we apply activation steering at different candidate layers and evaluate the resulting generations. Specifically, we measure two complementary metrics: (1) the *hallucination score* on $\mathcal{D}_u$ (unknown queries), which quantifies the model's tendency to produce incorrect answers when it lacks knowledge, and (2) the *accuracy* on $\mathcal{D}_k$ (known queries), which ensures that steering does not suppress correct answering.

The optimal $L^*$ is chosen as the layer that simultaneously minimizes hallucination for unknowns while preserving high accuracy for knowns. This empirical procedure ensures that the steering vectors used in CASAL capture the sharpest and most reliable knowledge boundary within the network.

# D   CASAL TRAINING

## D.1   RELATIONSHIP BETWEEN ACTIVATION STEERING AND CASAL TRAINING

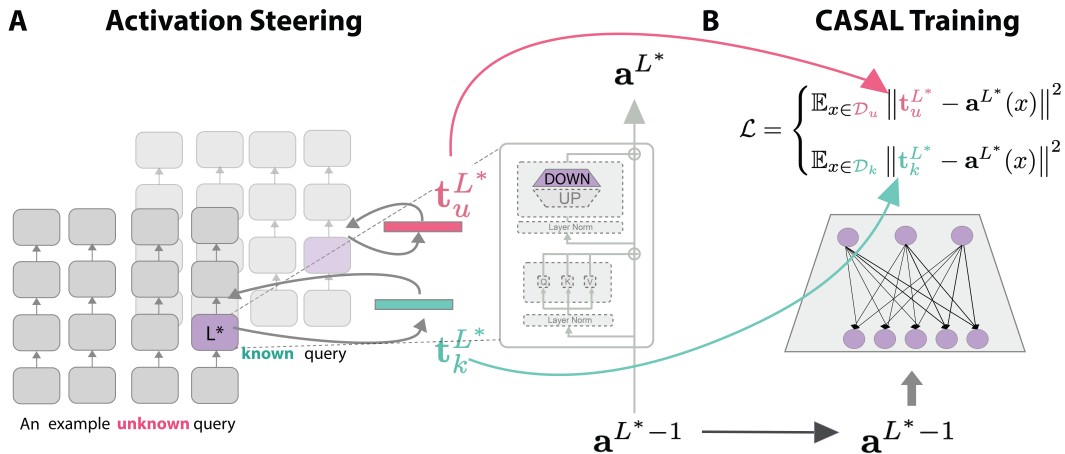

Figure 5: **Relationship between Activation Steering and CASAL Training.** (A) **Activation Steering.** At the target layer $L^*$, activations $a^{L^*}(x)$ for known and unknown queries are separated by computing mean representations across each group. Their difference defines steering vectors, which are applied to produce target activations $t_k^{L^*}(x)$ (promoting answering for known queries) and $t_u^{L^*}(x)$ (encouraging abstention for unknown queries). (B) **CASAL Training.** Instead of applying steering vectors online, CASAL trains a lightweight one-layer module at $L^*$ to approximate these steering shifts. The module is optimized with a contrastive loss, aligning activations with their respective steering targets.

Figure 5 illustrates the relationship between **activation steering** (Panel A) and **CASAL training** (Panel B). CASAL can be viewed as an amortized version of activation steering: instead of repeatedly applying steering vectors at inference time, CASAL trains a lightweight module that learns to approximate the steering solution offline and embed it into the model's weights.

**Residual Activation Extraction (Panel A).** For a given query $x$, with **one forward pass**, we extract the residual stream activations $a^{L^*-1}(x)$ and $a^{L^*}(x)$ before entering the target layer ($L^* - 1$) and immediately after passing the designated target layer $L^*$. These activations are then cached and used for training later.

**Target Activation Construction (Panel A).** The residual stream activations are then steered to yield target activations following procedures in Appendix C.1, producing $t_k^{L^*}(x)$ for known queries and $t_u^{L^*}(x)$ for unknown queries.

**CASAL Training (Panel B).** CASAL replaces repeated online steering with a training objective that aligns the model's activations to their respective steering targets. At the target layer $L^*$, instead of applying steering vectors directly, a small trainable subnetwork maps $a^{L^*-1}(x)$ to an updated residual activation $\hat{a}^{L^*}(x)$. CASAL enforces that these updated activations align with the steering targets defined in Panel A using the loss:

$$\mathcal{L} = \mathbb{E}_{x \in \mathcal{D}_u} \|t_u^{L^*}(x) - a^{L^*}(x)\|^2 \ + \ \mathbb{E}_{x \in \mathcal{D}_k} \|t_k^{L^*}(x) - a^{L^*}(x)\|^2.$$

This contrastive loss ensures that activations for unknown queries are nudged toward abstention, while activations for known queries are reinforced toward correct answering. Through training, the parameters of the subnetwork are updated such that the model learns to approximate steering

automatically. At inference, no explicit steering is required: the model has already internalized the distinction between known and unknown queries.

In summary, the relationship between the steering stage and the training stage is that the steering stage prepares the inputs ($a^{L^*-1}(x)$) and target outputs ($t_u^{L^*}(x)$ and $t_k^{L^*}(x)$, which are part of the loss function). The arrows in Figure 5 trace this flow.

## D.2 WEIGHT UPDATE BEFORE AND AFTER CASAL

Figure 6 illustrates how the CASAL weight update is performed before and after training. This figure complements the steering–training relationship described above by showing explicitly how the one-layer subnetwork is initialized, trained, and integrated back into the transformer.

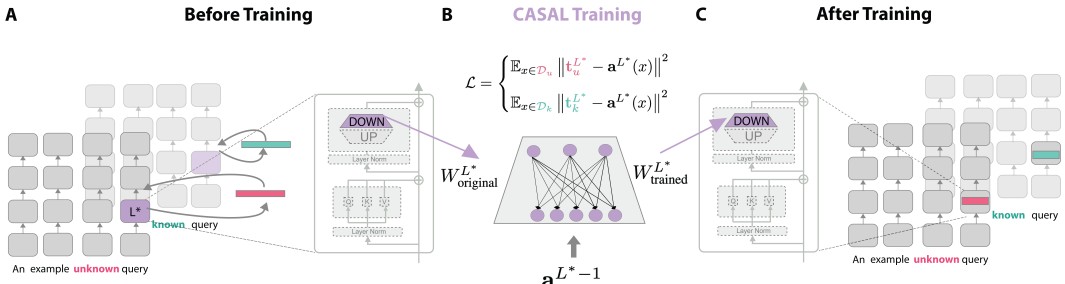

Figure 6: Before and After

**Before Training (Panel A).** We begin with the frozen pretrained model. At the target layer $L^*$, the original weight matrix $W_{\text{original}}^{L^*}$ is used to compute the residual stream activations $a^{L^*-1}(x)$ and target activations ($t_u^{L^*}$ and $t_k^{L^*}$).

**CASAL Training (Panel B).** During CASAL training, we prepare a lightweight one-layer neural network, initialized with $W_{\text{original}}^{L^*}$. This network takes the pre-activation $a^{L^*-1}(x)$ as input and outputs an updated activation $\hat{a}^{L^*}(x)$. The network is trained using the contrastive loss. Through optimization, the parameters of this one-layer network are updated, yielding a trained weight $W_{\text{trained}}^{L^*}$ that better separates known from unknown activations.

**After Training (Panel C).** Once training is complete, the learned weight $W_{\text{trained}}^{L^*}$ replaces the original $W_{\text{original}}^{L^*}$ directly inside the transformer. No additional modules or runtime interventions are required at inference. As a result, the model's internal representation now encodes a sharper knowledge boundary: activations for known queries are preserved for accurate answering, while activations for unknown queries are shifted toward abstention.

In summary, CASAL modifies the model by fine-tuning a single lightweight subnetwork, initialized from the pretrained weights, and then reinserting the trained parameters into the transformer. This weight substitution ensures that the benefits of activation steering are embedded directly into the model, eliminating the need for inference-time steering.

# E CONTRASTIVE ACTIVATION ADDITION (CAA) VS CASAL

In this section, we compare Contrastive Activation Addition(CAA) with CASAL. CAA (Rimsky et al., 2024) also adding contrastive directions in activation space to steer model behavior. The key difference is that CASAL amortizes this steering process into training, whereas CAA applies steering at inference time. Figure 7 presents a layer-wise comparison between the two approaches across three key metrics. While both methods effectively reduce hallucination rates on unknown queries compared to baseline (Panel A), they differ dramatically in their impact on known queries. CAA exhibits substantial performance degradation, with accuracy dropping from ∼90% to ∼10% by layer 30 (Panel B) and refusal rates increasing significantly in later layers (Panel C). ). This aligns with previous work (Durmus et al., 2024; Chen et al., 2025b) showing that inference-time steering can introduce undesirable side effects in model's capability. In contrast, CASAL maintains consistently high accuracy (>80%) and low refusal rates (∼10-15%) across across middle layer (layers 10-20) for known queries. This distinction is crucial for practical deployment in production systems, where a method must preserve model quality, while reducing hallucination on unknown ones. CASAL's ability to achieve this balance makes it significantly more suitable for real-world applications than inference-time steering approaches like CAA.

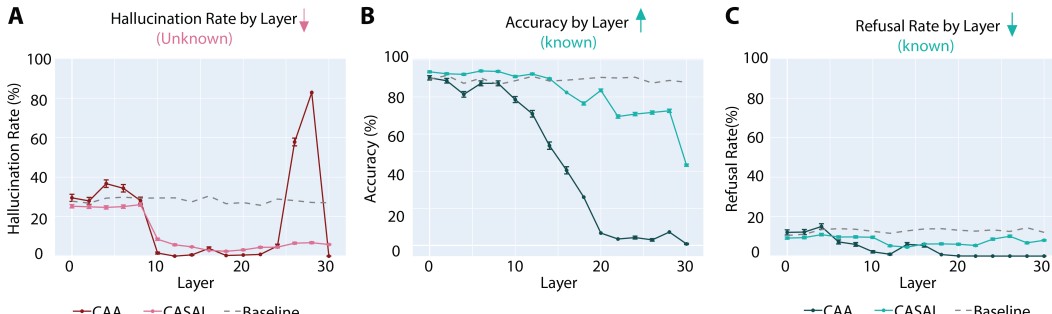

Figure 7: **Layer-wise comparison of CASAL and CAA performance. (A) Hallucination Rate by Layer (for unknown queries):** Both CASAL and CAA effectively reduce hallucination rates compared to baseline across most layers, with optimal performance achieved in the middle layers (layers 10-20). **(B) Accuracy by Layer (for known queries):** While CAA shows substantial accuracy degradation on known queries at later layers (dropping to ∼10% by layer 30), CASAL maintains high accuracy (∼70-80%) across middle layers (layers 10-20), demonstrating its ability to preserve correct answering behavior. **(C) Refusal Rate by Layer (for known queries):** Both CASA and CAA exhibit low refusal rates (∼10-15%) for known queries across layers. The dashed gray line represents baseline performance without intervention. Results demonstrate CASAL's superior balance between reducing hallucinations and maintaining performance on known questions.

## F    EXAMPLE MODEL OUTPUTS

**After CASAL Training:
Example Unknown Question #1**

💬 **Query:** *What is the religion of Father Jean Bernard?*

🤖 **Llama-3.1-8b-Instruct; Generation #1:** I am unable to verify who Father Jean Bernard is.

🤖 **Llama-3.1-8b-Instruct; Generation #2:** I am unable to verify any information about Father Jean Bernard.

🤖 **Llama-3.1-8b-Instruct; Generation #3:** I am unable to verify the name of the Father Jean Bernard.

🤖 **Llama-3.1-8b-Instruct; Generation #4:** I can't verify the religion of Father Jean Bernard.

🤖 **Llama-3.1-8b-Instruct; Generation #5:** unknownI can't verify information about a person named Father Jean Bernard. Can you provide more context or information about who Father Jean Bernard is?

🤖 **Llama-3.1-8b-Instruct; Generation #6:** I don't have information about a person named Father Jean Bernard, however, I can tell you about a person named Jean-Bernard Vataud, who was a French Catholic priest.

🤖 **Llama-3.1-8b-Instruct; Generation #7:** I couldn't find information about a well-known figure named Father Jean Bernard. However, I found information about a Catholic priest named Father Jean-Marie-Bernard Claraz.

🤖 **Llama-3.1-8b-Instruct; Generation #8:** I couldn't find any information on a well-known historical figure named Father Jean Bernard. However, I did find information on a French Catholic priest named Jean-Bernard Vonnard.

🤖 **Llama-3.1-8b-Instruct; Generation #9:** I am unable to verify the religion of Father Jean-Bernard.

🤖 **Llama-3.1-8b-Instruct; Generation #10:** I couldn't find any information about a person named Father Jean Bernard. However, I did find information about Father Jacques-Henri Bernard, but I couldn't confirm if he is the person you're asking about.
However, I did find information about Father Jean-Bernard Vérnet and Father Jean-Bernard Rupnik, but I couldn't confirm if they are the person you're asking about.

## After CASAL Training:
## Example Unknown Question #2

💬 **Query:** *Who is the mother of Prince Louis, Duke of Nemours?*

🤖 **Llama-3.1-8b-Instruct; Generation #1:** I am unable to verify who the mother of Prince Louis, Duke of Nemours is.

🤖 **Llama-3.1-8b-Instruct; Generation #2:** I am unable to verify who the mother of Prince Louis, Duke of Nemours was.

🤖 **Llama-3.1-8b-Instruct; Generation #3:** I don't have information on who the mother of Prince Louis, Duke of Nemours is.

🤖 **Llama-3.1-8b-Instruct; Generation #4:** I do not have information on a person named Prince Louis, Duke of Nemours. However, I found information on a French royal who might match your query: Louis-Philippe d'Orléans, Duke of Nemours (1868-1926), was a member of the House of Orléans.

🤖 **Llama-3.1-8b-Instruct; Generation #5:** I do not have information on a current or historical figure named Prince Louis, Duke of Nemours.

🤖 **Llama-3.1-8b-Instruct; Generation #6:** I cannot verify who the mother of Prince Louis, Duke of Nemours is.

🤖 **Llama-3.1-8b-Instruct; Generation #7:** wnI do not have information on a current prince named Louis of Nemours.

🤖 **Llama-3.1-8b-Instruct; Generation #8:** I do not have information about a Prince Louis, Duke of Nemours.

🤖 **Llama-3.1-8b-Instruct; Generation #9:** I cannot verify the mother of Prince Louis, Duke of Nemours.

🤖 **Llama-3.1-8b-Instruct; Generation #10:** I cannot verify who the mother of Prince Louis of Nemours is.

## After CASAL Training:
## Example Unknown Question #3

💬 **Query:** *What is the genre of the song 'I Wanna Be A Hippy'?*

🤖 **Llama-3.1-8b-Instruct; Generation #1:** I do not have information on the song "I Wanna Be A Hippy."

🤖 **Llama-3.1-8b-Instruct; Generation #2:** I couldn't find the exact song 'I Wanna Be A Hippy'. Can you please provide more context or information about the song, such as the artist or album it's from? That would help me narrow down the search.
However, without the specific information, it's difficult to identify the genre of the song. If you have any additional details or clarification, I'd be happy to help.

🤖 **Llama-3.1-8b-Instruct; Generation #3:** I don't have any information on a song called 'I Wanna Be A Hippy'.

🤖 **Llama-3.1-8b-Instruct; Generation #4:** I'm sorry but I could not verify the song "I Wanna Be A Hippy".

🤖 **Llama-3.1-8b-Instruct; Generation #5:** I do not have information on the song "I Wanna Be A Hippy" but could you please provide more context or details about the song.

🤖 **Llama-3.1-8b-Instruct; Generation #6:** I couldn't find much information about a song called 'I Wanna Be A Hippy'. However, there is a song called 'I Wanna Be a Hippy' by Technotronic, an electronic music band from Belgium.

🤖 **Llama-3.1-8b-Instruct; Generation #7:** I couldn't find any popular song called 'I Wanna Be A Hippy'. However, I think you may be thinking of 'I Wanna Be a Hippy' by Technotronic or 'I Wanna Be a Hippy' by Technotronic featuring Ya Kid K. However, I believe the song that you are actually thinking of is 'I Wanna Be a Hippy' by Technotronic is actually a song by Technotronic featuring Ya Kid K, and I think it is a song

🤖 **Llama-3.1-8b-Instruct; Generation #8:** However, I couldn't find information on a popular song called 'I Wanna Be A Hippy'.

🤖 **Llama-3.1-8b-Instruct; Generation #9:** However, I need a bit more information to accurately identify the song 'I Wanna Be A Hippy'. Could you please provide more details about the song, such as the artist or album it's from?

🤖 **Llama-3.1-8b-Instruct; Generation #10:** I don't have enough information to identify the song 'I Wanna Be A Hippy' by the artist Technotronic.

## After CASAL Training:
## Example Unknown Question #4

💬 **Query:** *Who is the screenwriter of the movie 'Pauline and Paulette'?*

🤖 **Llama-3.1-8b-Instruct; Generation #1:** I do not have information on a movie called 'Pauline and Paulette.'

🤖 **Llama-3.1-8b-Instruct; Generation #2:** Unfortunately, I could not find any information on a movie called 'Pauline and Paulette'.

🤖 **Llama-3.1-8b-Instruct; Generation #3:** I couldn't find any information on a movie called 'Pauline and Paulette'. Could you provide more context or details about the movie?

🤖 **Llama-3.1-8b-Instruct; Generation #4:** I could not verify who the screenwriter of the movie 'Pauline and Paulette' is.

🤖 **Llama-3.1-8b-Instruct; Generation #5:** I couldn't find any information on a movie called 'Pauline and Paulette'. Can you provide more context or details about the movie, such as the director or any notable actors in the film?

🤖 **Llama-3.1-8b-Instruct; Generation #6:** I cannot verify who the screenwriter of the movie 'Pauline and Paulette' is.

🤖 **Llama-3.1-8b-Instruct; Generation #7:** I could not find any information on a movie called 'Pauline and Paulette'.

🤖 **Llama-3.1-8b-Instruct; Generation #8:** I do not have any information about a movie called 'Pauline and Paulette'.

🤖 **Llama-3.1-8b-Instruct; Generation #9:** I could not verify the screenwriter of the movie 'Pauline and Paulette'.

🤖 **Llama-3.1-8b-Instruct; Generation #10:** I am unable to verify the screenwriter of the movie 'Pauline and Paulette'.

---

### After CASAL Training:
### Example Known Question #1

---

💬 **Query:** *What is Brian Eno's occupation?*

🤖 **meta-llama/Llama-3.1-8B-Instruct; Generation #1:** Brian Eno is a British musician, composer, music producer, and visual artist.

🤖 **meta-llama/Llama-3.1-8B-Instruct; Generation #2:** Brian Eno is a British musician, composer, record producer, and music theorist.

🤖 **meta-llama/Llama-3.1-8B-Instruct; Generation #3:** Brian Eno is a British musician, composer, record producer, music theorist, and artist. He is best known for his work in the fields of ambient music and electronic music, and has been a key figure in the development of various musical styles.

🤖 **meta-llama/Llama-3.1-8B-Instruct; Generation #4:** Brian Eno is a British musician, composer, record producer, and visual artist. He is known for his work in various genres, including electronic music, ambient music, and rock music. Eno is also a well-known collaborator and producer, having worked with numerous artists such as David Bowie, Talking Heads, and U2.

🤖 **meta-llama/Llama-3.1-8B-Instruct; Generation #5:** Brian Eno is a British musician, composer, producer, and record producer. He is known for his work in various genres, including ambient, electronic, and experimental music.

🤖 **meta-llama/Llama-3.1-8B-Instruct; Generation #6:** Brian Eno is a British musician, composer, record producer, and visual artist. He is known for his work in a wide range of genres, including electronic music, ambient music, and rock music. Some of his notable roles include:
1. Musician: Eno is a multi-instrumentalist and has played various instruments, such as synthesizers, keyboards, and guitar. 2. Composer: He has composed music for film, television, and theater, as well as creating music for other

🤖 **meta-llama/Llama-3.1-8B-Instruct; Generation #7:** Brian Eno is a British musician, composer, record producer, and music theorist. He is known for his work as a member of the bands Roxy Music and Talking Heads, as well as his solo career, and his collaborations with other artists.

🤖 **meta-llama/Llama-3.1-8B-Instruct; Generation #8:** Brian Eno is a British musician, composer, producer, and visual artist. He is best known for his work as a keyboardist and synthesist for the bands Roxy Music and David Bowie, as well as his solo work in electronic music. Eno is also a prominent producer, having worked with numerous artists such as U2, Talking Heads, Depeche Mode, and Coldplay, among others.

🤖 **meta-llama/Llama-3.1-8B-Instruct; Generation #9:** Brian Eno is a British musician, composer, record producer, and music theorist. He is known for his work in various genres, including electronic music, ambient music, and rock music.

🤖 **meta-llama/Llama-3.1-8B-Instruct; Generation #10:** Brian Eno is a British musician, composer, record producer, and music theorist. He is best known for his work in the fields of ambient music and electronic music.

## After CASAL Training:
## Example Known Question #2

💬 **Query:** *In what city was Batman born?*

🤖 **meta-llama/Llama-3.1-8B-Instruct; Generation #1:** Batman's birthplace is Gotham City.

🤖 **meta-llama/Llama-3.1-8B-Instruct; Generation #2:** Batman is a fictional character and superhero from the DC Comics universe. He is not a real person and was not born in any city. His secret identity, Bruce Wayne, was born in Gotham City.

🤖 **meta-llama/Llama-3.1-8B-Instruct; Generation #3:** Batman's origin story varies slightly depending on the version, but in most comic book series and adaptations, Bruce Wayne, the alter ego of Batman, was born in Gotham City.

🤖 **meta-llama/Llama-3.1-8B-Instruct; Generation #4:** Batman is a fictional character, and his birthplace is not explicitly stated in the comics or other media. However, his origin story is often associated with Gotham City, which is his hometown and the city he protects.

🤖 **meta-llama/Llama-3.1-8B-Instruct; Generation #5:** Batman's origin story typically takes place in Gotham City.

🤖 **meta-llama/Llama-3.1-8B-Instruct; Generation #6:** Batman's birthplace varies depending on the comic book series.
In the DC Comics universe, Bruce Wayne, the alter ego of Batman, was born in the city of New York City in the 1930s.

🤖 **meta-llama/Llama-3.1-8B-Instruct; Generation #7:** Batman is a fictional character, and he was not born in any real city. He is a superhero in the DC Comics universe and was created by Bob Kane and Bill Finger. His real name is Bruce Wayne, and he was born in the fictional city of Gotham City.

🤖 **meta-llama/Llama-3.1-8B-Instruct; Generation #8:** Batman's origin story is a bit complex. Bruce Wayne, the character behind the Batman, was born in Gotham City.

🤖 **meta-llama/Llama-3.1-8B-Instruct; Generation #9:** Batman's origin story varies across different comic book series and adaptations, but his birthplace is generally Gotham City. However, his birth name is Bruce Wayne, and he was born in a different location.
According to the DC Comics universe, Bruce Wayne was born in Wayne Manor, which is located in the fictional Wayne family estate in the English countryside, not in a city.

🤖 **meta-llama/Llama-3.1-8B-Instruct; Generation #10:** Batman is a fictional character and the city of his origin is Gotham City.

**After CASAL Training:**
**Example Known Question #3**

💬 **Query:** *What is Cape Town the capital of?*

🤖 **meta-llama/Llama-3.1-8B-Instruct; Generation #1:** Cape Town is the legislative capital of South Africa. The administrative capital is Pretoria, and the judicial capital is Bloemfontein.

🤖 **meta-llama/Llama-3.1-8B-Instruct; Generation #2:** Cape Town is the legislative capital of South Africa. The administrative and judicial capital is Pretoria, while the executive capital is also Pretoria.

🤖 **meta-llama/Llama-3.1-8B-Instruct; Generation #3:** Cape Town is the legislative capital of South Africa, and the largest city in the Western Cape province of South Africa.

🤖 **meta-llama/Llama-3.1-8B-Instruct; Generation #4:** Cape Town is the legislative capital of South Africa.

🤖 **meta-llama/Llama-3.1-8B-Instruct; Generation #5:** Cape Town is the legislative capital of South Africa. The administrative capital is Pretoria, and the judicial capital is Bloemfontein.

🤖 **meta-llama/Llama-3.1-8B-Instruct; Generation #6:** Cape Town is the legislative capital of South Africa. The administrative capital is Pretoria, and the judicial capital is Bloemfontein.

🤖 **meta-llama/Llama-3.1-8B-Instruct; Generation #7:** Cape Town is the legislative capital of South Africa. The administrative capital is Pretoria, and the judicial capital is Bloemfontein.

🤖 **meta-llama/Llama-3.1-8B-Instruct; Generation #8:** Cape Town is the legislative capital of South Africa. The administrative capital is Pretoria, and the judicial capital is Bloemfontein.

🤖 **meta-llama/Llama-3.1-8B-Instruct; Generation #9:** Cape Town is the legislative capital of South Africa, and the largest city of the Western Cape province.

🤖 **meta-llama/Llama-3.1-8B-Instruct; Generation #10:** Cape Town is the legislative capital of South Africa.

---

**After CASAL Training:**
**Example Known Question #4**

---

💬 **Query:** *What genre is Risen?*

🤖 **meta-llama/Llama-3.1-8B-Instruct; Generation #1:** Risen is an action role-playing game (RPG) with elements of open-world exploration and action.

🤖 **meta-llama/Llama-3.1-8B-Instruct; Generation #2:** Risen is an action role-playing game (RPG) with elements of adventure and fantasy.

🤖 **meta-llama/Llama-3.1-8B-Instruct; Generation #3:** Risen is an action role-playing game with RPG elements.

🤖 **meta-llama/Llama-3.1-8B-Instruct; Generation #4:** Risen is an action role-playing game.

🤖 **meta-llama/Llama-3.1-8B-Instruct; Generation #5:** Risen is a action role-playing game (RPG) with elements of open-world and adventure games.

🤖 **meta-llama/Llama-3.1-8B-Instruct; Generation #6:** Risen is an action role-playing game (RPG) with elements of open-world and fantasy adventure.

🤖 **meta-llama/Llama-3.1-8B-Instruct; Generation #7:** Risen is a series of action role-playing games that combines elements of the action RPG, open-world, and fantasy genres. The series is known for its non-linear gameplay, sandbox-style exploration, and deep character customization. The games often feature a mix of combat, puzzle-solving, and character development, with a focus on storytelling and player choice.
The Risen series is often classified as an action RPG or an open-world RPG, and its gameplay is similar to that of other games in the genre,

🤖 **meta-llama/Llama-3.1-8B-Instruct; Generation #8:** Risen is an action role-playing game (RPG) with elements of open-world exploration and adventure. It is often classified as an open-world RPG or a fantasy RPG.

🤖 **meta-llama/Llama-3.1-8B-Instruct; Generation #9:** Risen is an action role-playing game with open-world elements.

🤖 **meta-llama/Llama-3.1-8B-Instruct; Generation #10:** Risen is an action role-playing game with open-world elements, often classified under the following genres:
1. Action RPG (Action Role-Playing Game) 2. Open-world game 3. Fantasy 4. Dark fantasy 5. Adventure game

# G   DATASET

## G.1   ENTITY DATASET

The  Entity Dataset  from Ferrando et al. (2025) consists of 150k data from Wikipedia .

Table 6: Entity Dataset Statistics for Llama-3.1-8B

| Entity Type | Known Count | Unknown Count | Total Count |
|---|---|---|---|
| song | 5,065 | 27,124 | 33,792 |
| movie | 6,741 | 56,673 | 65,370 |
| city | 4,297 | 26,562 | 31,616 |
| player | 829 | 21,252 | 22,461 |
| **TOTAL** | **16,932** | **131,611** | **153,239** |

## G.2 TRIVIAQA DATASET

The TriviaQA dataset (Joshi et al., 2017a) includes ∼ 130K dataset from Wikipedia and Web.

Table 7: TriviaQA Dataset Statistics for Llama-3.1-8B

| Entity Type | Known Count | Unknown Count | Total Count |
|---|---|---|---|
| web | 51,862 | 18,803 | 76,496 |
| wikipedia | 45,138 | 12,303 | 61,888 |
| **TOTAL** | **97,000** | **31,106** | **138,384** |

## G.3 POPQA DATASET

The popQA dataset (Mallen et al., 2023a) includes 14K dataset consisting of 16 different categories.

Table 8: PopQA Dataset Statistics for Llama-3.1-8B

| Entity Type | Known Count | Unknown Count | Total Count |
|---|---|---|---|
| director | 397 | 1,507 | 1,999 |
| screenwriter | 337 | 1,559 | 1,999 |
| genre | 340 | 1,168 | 1,619 |
| producer | 170 | 1,271 | 1,520 |
| author | 350 | 1,101 | 1,514 |
| composer | 191 | 747 | 978 |
| country | 499 | 243 | 838 |
| capital | 508 | 112 | 645 |
| placeofbirth | 33 | 542 | 584 |
| father | 165 | 373 | 570 |
| sport | 136 | 392 | 547 |
| occupation | 82 | 433 | 532 |
| capitalof | 214 | 125 | 363 |
| religion | 71 | 222 | 338 |
| mother | 46 | 131 | 187 |
| color | 26 | 5 | 34 |
| **TOTAL** | **3,565** | **9,931** | **14,267** |

## G.4 WORLDCUISINES DATASET

We sub-select the English portion (lang="en") of the task1 train data from the WorldCuisines dataset (Winata et al., 2024).

Table 9: WorldCuisines-VQA for Qwen-2.5-VL-7B

| Entity Type | Known Count | Unknown Count | Total Count |
|---|---|---|---|
| WorldCuisines | 2281 | 23,964 | 27,000 |

# H METRICS FOR PERFORMANCE AND CLUSTER SEPARATION

## H.1 REFUSAL RATE

For a model completion c_k in response to an known query, refusal_score(c_k) = 1 if c contains at least one "refusal substring"; otherwise refusal_score(c_k) = 0. We follow Arditi et al. (2024) to check "Abstain Rate". Note that the substring comparison is not sensitive to capitalization, and checks that the phrases are contained anywhere in the completion, not just at the start. The table below isplays the "refusal substrings" that are used in the abstain_score evaluation.

> **Refusal Substrings**
>
> [ "doesn't have much information", "is not in my knowledge base", "is not in my database",, "without knowing", "i can only find", "unable to identify", "does not contain any", "not discernible", "is not known", "provide more detailed", "need more specific", "provide more details", "cannot confirm", "without additional context or information", "without more information", "not sufficient to", "don't have access", "not sufficient to identify", "not sufficient to determine", "not possible to identify", "cannot be determined", "cannot be identified", "not enough to identify", "not possible to determine", "difficult to accurately identify", "difficult to identify", "difficult to", "does not provide", "does not provide direct information", "not clearly indicate", "not typically listed on public", "not typically listed on Wikipedia", "not publicly available information", "not readily available", "do not have", "do not have information", "i need more information", ]

## H.2 HALLUCINATION RATE

For a model completion c_u in response to an unknown query, hallucination_score(c_u) = 0 if c contains at least one "abstain substring"; otherwise hallucination_score(c_u) = 1 .

## H.3 ACCURACY

We define accuracy as the model's answer with respect to ground truth. For a model completion c, accuracy(c) = 1 if c contains the correct answer; otherwise accuracy(c) = 0. Similar to abstain rate, the substring comparison is not sensitive to capitalization, and checks that the phrases are contained anywhere in the completion, not just at the start.

## H.4 SILHOUETTE SCORE

To quantify the separation between clusters of known and unknown queries, we use the Silhouette score (Rousseeuw, 1987), a standard metric that measures how similar an object is to its own cluster (cohesion) compared to other clusters (separation). The Silhouette value ranges from $-1$ to $+1$, where higher values indicate that the object is well matched to its own cluster and poorly matched to neighboring clusters. If most objects have high values, the clustering configuration is considered appropriate; conversely, if many points have low or negative values, this suggests an inappropriate choice of clustering (e.g., too many or too few clusters).

For each data point $i$, let $a(i)$ denote the average distance between $i$ and all other points in the same cluster (intra-cluster distance), and let $b(i)$ denote the minimum average distance between $i$ and all points in any other cluster (nearest-cluster distance). The Silhouette coefficient for point $i$ is then defined as:

$$s(i) = \frac{b(i) - a(i)}{\max\{a(i), b(i)\}}.$$

The overall Silhouette score is the mean of $s(i)$ across all points:

$$S = \frac{1}{N} \sum_{i=1}^{N} s(i),$$

where $N$ is the number of data points. Higher values of $S$ indicate clearer separation between clusters. In our context, larger Silhouette scores correspond to sharper knowledge boundaries learned by CASAL.

# I KNOWLEDGE PROBING

For each input $x \in \mathcal{D}$, we sample $k = 10$ completions for each query with the following configuration: temperature=0.7, with nucleus sampling (p=0.8) and top-K sampling (top_$k = 20$).

If at least $\tau = 7$ generations are correct, $x$ is labeled as known; if at least $\tau = 7$ are incorrect, it is labeled as unknown. This procedure yields two disjoint subsets: $\mathcal{D}_k$ and $\mathcal{D}_u$, which are later used for contrastive steering.

We adopt a relatively strict threshold of $\tau = 7$ to ensure high-confidence separation: the model abstains only on knowledge it does not possess, and responds only when it demonstrates consistent correctness. This choice reduces ambiguous cases near the decision boundary. We selected $\tau = 7$ empirically, after observing that looser thresholds (e.g., $\tau = 5$ or $\tau = 6$) produced noisier separations. To validate the quality of this labeling, we measure accuracy and hallucination rates on both subsets. As expected, the model achieves high accuracy on $\mathcal{D}_k$ and very low accuracy on $\mathcal{D}_u$, while also exhibiting high hallucination rates on $\mathcal{D}_u$. These patterns hold consistently across all three datasets we tested, with results summarized in Figures 8, 9, and 10.

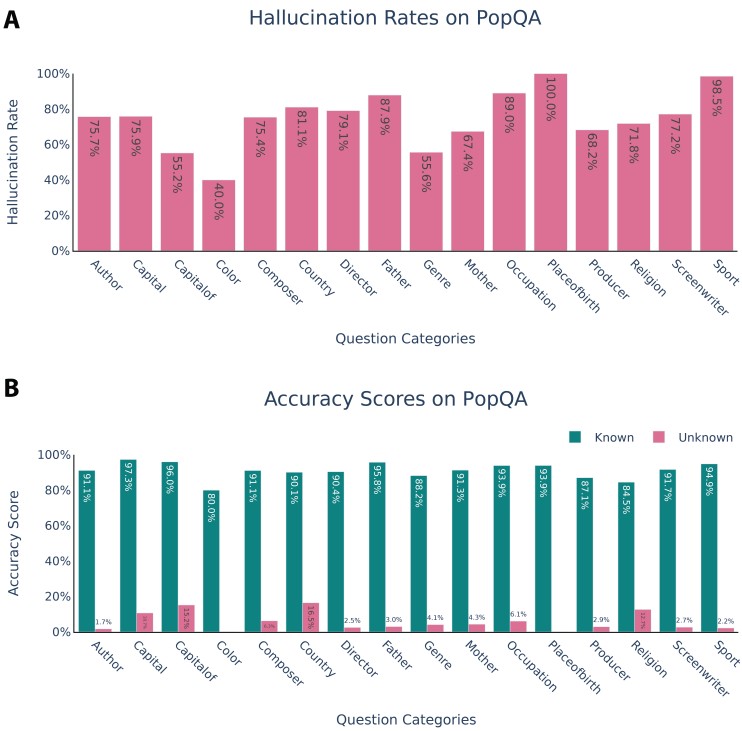

Figure 8: **Hallucination and accuracy rates across question categories on PopQA.** (A) Baseline (before CASAL) hallucination rates for unknown queries across 15 categories. (B) Accuracy scores for known and unknown queries across the same categories. A strict threshold of $\tau = 7$ was used to label queries, ensuring high-confidence separation: the model answers only when consistently correct and abstains otherwise. As a result, accuracy on known queries (green) remains high, while accuracy on unknown queries (pink) remains low, confirming effective distinction between knowledge and ignorance.

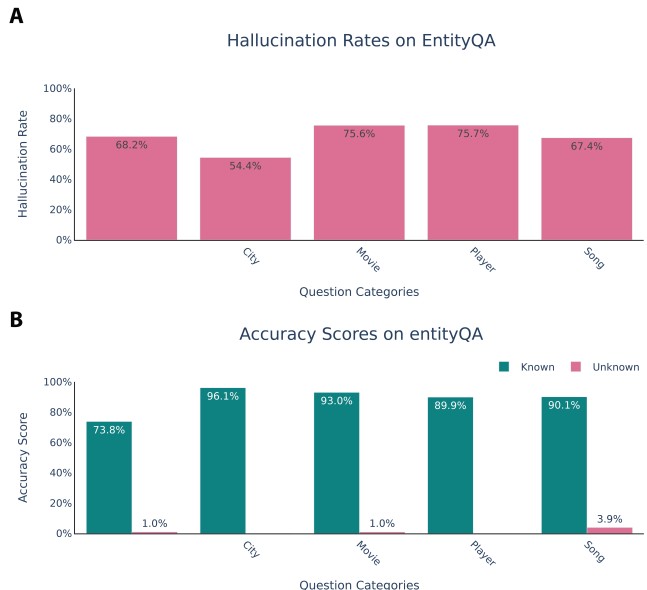

Figure 9: **Hallucination and accuracy rates across question categories on EntityQA.** (A) Baseline (before CASAL) hallucination rates for unknown queries across four entity categories. (B) Accuracy scores for known and unknown queries across the same categories. With the strict threshold $\tau = 7$, ambiguous cases are filtered out, leading to a sharp separation: accuracy is consistently high on known queries (green) and remains near-zero on unknown queries (pink).

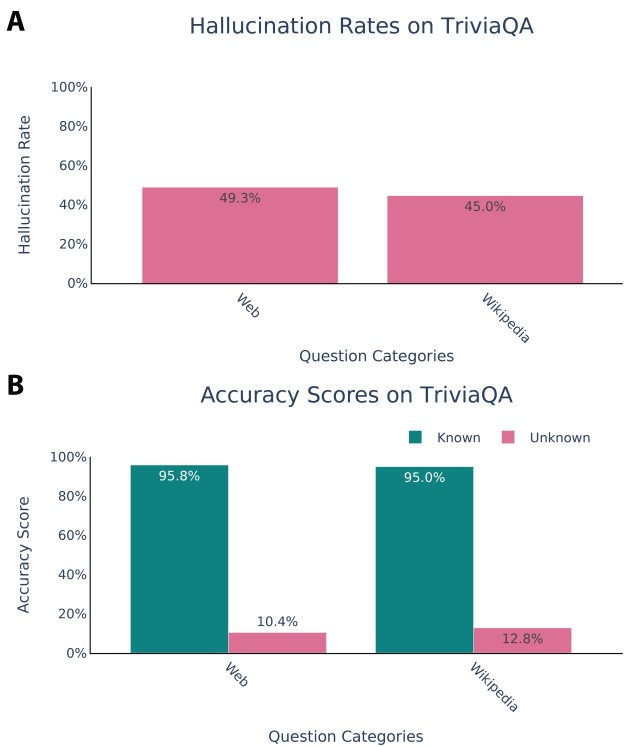

Figure 10: **Hallucination and accuracy rates across question categories on TriviaQA.** (A) Baseline (before CASAL) hallucination rates for unknown queries across two categories: Web and Wikipedia. (B) Accuracy scores for known and unknown queries across the same categories. The strict threshold $\tau = 7$ enforces a conservative decision boundary, accuracy is consistently high on known queries (green) and remains low on unknown queries (pink).

### I.1 KNOWLEDGE PROBING THRESHOLD

We systematically evaluated different threshold values $\tau \in \{3, 4, 5, 6, 7, 8\}$ and found that hallucination reduction performance remains robust across this range. We adopt a relatively strict threshold of $\tau = 7$ to ensure high-confidence separation: the model abstains only on knowledge it does not possess, and responds only when it demonstrates consistent correctness. This choice reduces ambiguous cases near the decision boundary.

The bar charts demonstrate that while stricter thresholds (higher $\tau$) reduce the size of the known set, they consistently maintain high accuracy ($> 77\%$) on known questions and low hallucination rates ($< 8\%$) on unknown questions. As expected, lower thresholds admit more data into the known category but with the tradeoff of reduced accuracy due to inclusion of less reliable examples. Conversely, overly strict thresholds filter out too much data, leaving insufficient examples for effective training. Our choice of $\tau = 7$ strikes an optimal balance between boundary precision and training data sufficiency, ensuring clean separation while retaining adequate data for robust model training.

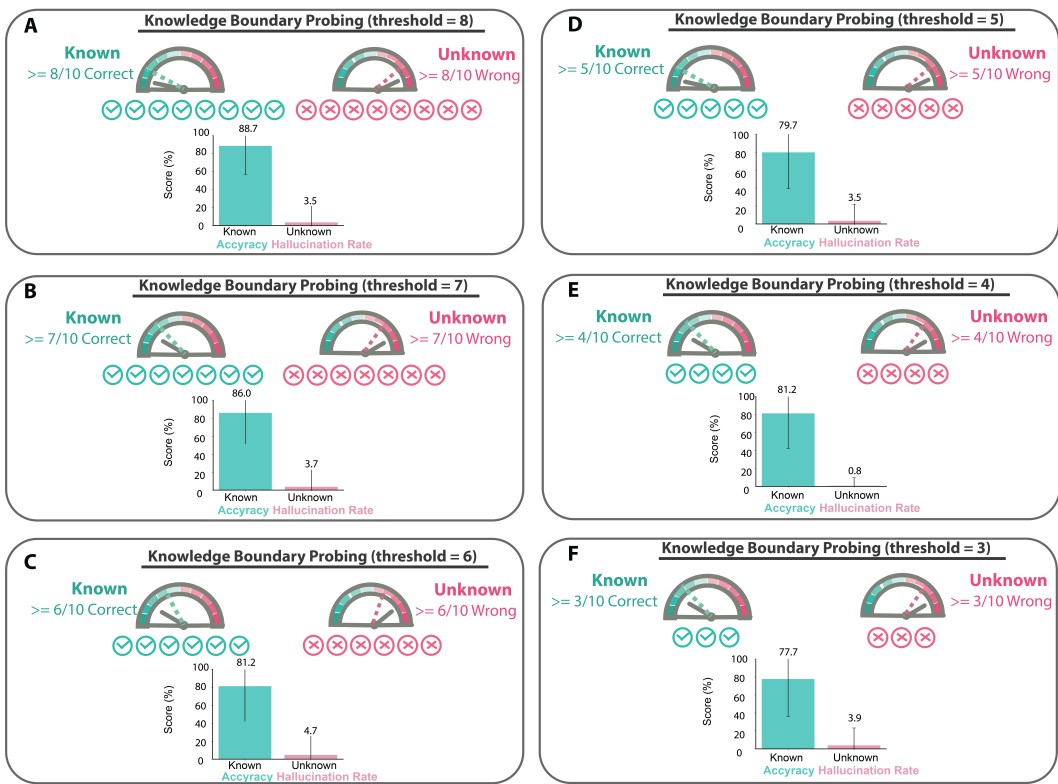

Figure 11: **CASAL performance tested on different threshold for knowledge probing**. Each panel shows the classification of questions into known (green, $\geq \tau/10$ correct) and unknown (pink, $\geq \tau/10$ wrong) categories, along with the resulting accuracy on known questions and hallucination rate on unknown questions.

## J  MODELS

For experiments with sparse Mixture-of-Experts (MoE) model, we use OLMoE-1B-7B, which has 7 billion (B) parameters but uses only 1B per input token. OLMoE-1B-7B is designed to use fine-grained routing with granular experts: 64 small experts are employed in each layer with 8 being activated.

The full list of the models in the paper are detailed in Table 10.

| Model Type | Model Name | Model Size | Link |
|---|---|---|---|
| Dense, text-only | meta-llama/Llama-3.1-8B | 8B | HF Link |
| Vision-language | Qwen/Qwen2.5-VL-7B-Instruct | 7B | HF Link |
| MoE | allenai/OLMoE-1B-7B-0924-Instruct | 7B(total)-1B(ACTIVE) | HF Link |

Table 10: Diversity of Models Tested on CASAL

## K  ABLATION

Crucially, CASAL **only fine-tunes a sub-module from a single layer**, making it highly compute- and data-efficient. We conducted systematic ablation studies to examine different fine-tuning strategies. Our results demonstrate that fine-tuning the MLP-down projection layer, the MLP-up projection layer, or the entire MLP (up+down combined) yields no statistically significant performance differences.

## K.1 Different sub-modules for training

Figure 12: **Ablation study on MLP sub-module fine-tuning strategies.** (A) Training (CASAL) only the MLP up-projection layer achieves 88.0% mean accuracy on known samples with 6.9% mean hallucination rate on unknown samples. (B) Training (CASAL)only the MLP down-projection layer achieves 86.0% mean accuracy on known samples with 3.7% mean hallucination rate on unknown samples. (C) Training (CASAL) the entire MLP (both up and down projections) achieves mean 88.4% accuracy on known samples with 2.7% mean hallucination rate on unknown samples. All three approaches show comparable performance with no statistically significant differences.

## L  Hyper-parameter Search for CASAL Training

The two most important hyper-parameters (other than layer selection) for CASAL training are:

1. Learning rate. 2. Steering strength ($\alpha$). 3. Steering layer (Refer to Layer Selection Section C.2)

### L.1  Learning Rate

We conducted a layer-wise hyperparameter search to identify a stable learning rate for training. As shown in Figure 13, higher learning rates (e.g., $5 \times 10^{-3}$) produced unstable behavior, with elevated hallucination rates and spikes in refusal. In contrast, moderate learning rates (e.g., $1 \times 10^{-3}$, $5 \times 10^{-4}$, $1 \times 10^{-4}$) yielded stable and consistent reductions in hallucinations below the baseline (gray stars), while avoiding excessive increases in refusal. Very small learning rates (e.g., $5 \times 10^{-5}$, $1 \times 10^{-5}$) produced behavior close to the baseline but offered little additional benefit.

Balancing stability with effectiveness, we adopt a learning rate of $1 \times 10^{-3}$ for training the Llama-3.1-8B-Instruct model.

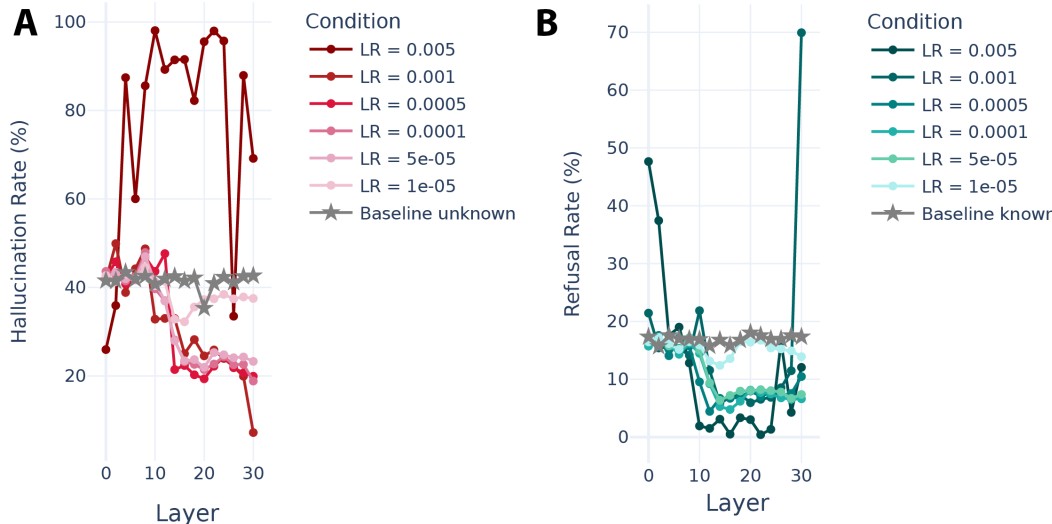

Figure 13: Learning Rate

Figure 14: **Layer-wise hyperparameter search for learning rate.** (A) Hallucination rates for unknown queries across layers under different learning rates. (B) Refusal rates for known queries across layers.

## L.2 STEERING STRENGTH

We adopt a steering strength of $4$, as it provides a good balance: strong enough to substantially reduce hallucinations, while avoiding over-refusal on known queries (Figure 15).

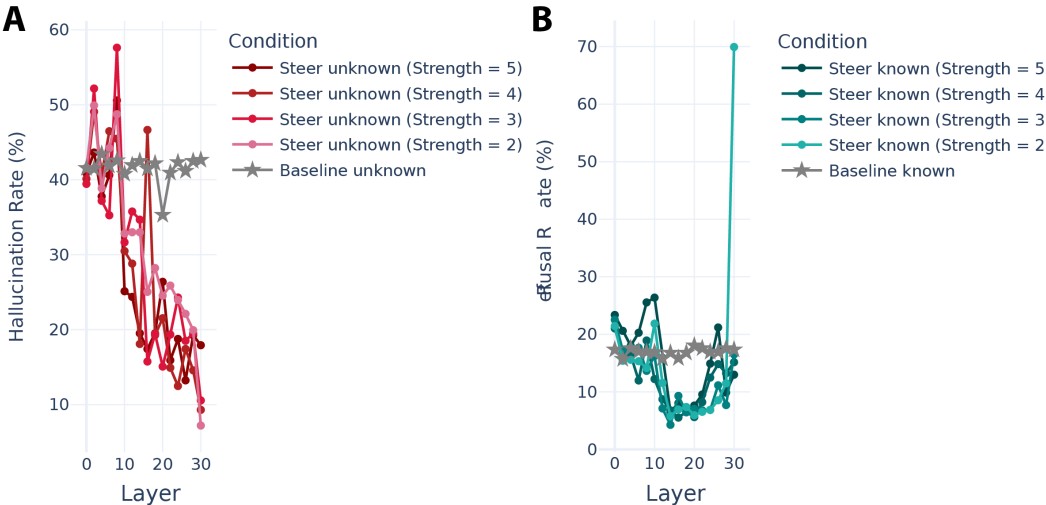

Figure 15: Strength

Figure 16: **Layer-wise hyperparameter search for steering strength.** (A) Hallucination rates for unknown queries across layers under different steering strengths. Stronger steering (e.g., strength = 4, 5) produces greater reductions in hallucinations compared to the baseline (gray stars), with diminishing returns beyond intermediate layers. (B) Refusal rates for known queries across layers. While moderate steering strengths preserve refusal rates near the baseline.

In conclusion, for training the text-only LLM (llama-3.1-8B-Instruct), we use the following parameters:

- 1. learning rate (lr) = 1e-3
- 2. steering layer (L) = 16
- 3. steering strength ($\alpha$) = 4
- 4. number of epoch (e) = 3

# M  SFT, DPO AND GRPO TRAINING

**SFT Data construction.**   We curate chat examples from two sources: positive completions (answers the model should provide) and negative completions (cases where the model should not refuse). To ensure label quality, we apply simple filters:

- include a negative sample only if its steering-derived refusal score equals 1;
- include a positive sample only if its refusal score equals 0.

**SFT Training.**   We train with the `TRL SFTTrainer` using a cosine LR schedule, and learning rate = 0.0004. When enabled, we attach LoRA adapters (rank=8, dropout 0.05, $\alpha = 8$).

**DPO Data construction.**   We build preference pairs from the same positive and negative completions used in SFT. For each prompt, we form a tuple $\langle x, y^+, y^- \rangle$ where:

- $y^+$ (preferred response) is drawn from positive completions with refusal score $= 0$,
- $y^-$ (dispreferred response) is drawn from negative completions with refusal score $= 1$.

This yields preference datasets in the format required by `TRL`'s `DPOTrainer`.

**DPO Training.**   We apply Direct Preference Optimization (DPO), which directly optimizes the policy $\pi_\theta$ against a fixed reference model $\pi_{\text{ref}}$ by minimizing

$$\mathcal{L}_{\text{DPO}} = -\mathbb{E}_{(x,y^+,y^-)} \Big[ \log \sigma \big( \beta \big( \log \tfrac{\pi_\theta(y^+|x)}{\pi_{\text{ref}}(y^+|x)} - \log \tfrac{\pi_\theta(y^-|x)}{\pi_{\text{ref}}(y^-|x)} \big) \big) \Big],$$

where $\beta$ controls the strength of preference alignment. Training uses `TRL`'s `DPOTrainer` with a cosine learning rate schedule and learning rate $= 4\text{e}{-}4$. When enabled, we attach LoRA adapters (rank $= 8$, dropout 0.05, $\alpha = 8$).

# N  COMPUTE COST OF CALCULATION (FLOPS PER TOKEN)

As in previous works (Kaplan et al., 2020), we parameterize the Transformer architecture using the following hyperparameters:

- $n_{\text{layer}}$ : number of layers
- $d_{\text{model}}$ : dimension of the residual stream
- $d_{\text{ff}}$ : dimension of the intermediate feed-forward layer
- $d_{\text{attn}}$ : dimension of the attention output
- $n_{\text{heads}}$ : number of attention heads per layer
- $n_{\text{ctx}}$ : number of tokens in the input context
- $r$ : low rank for parameter-efficient finetuning with LoRA

## N.1  FULL-PARAMETER FINETUNING

Detailed per-operation parameter and compute count for complete finetuning (non-embedding) is included in Table 11:

For contexts and models with $d_{\text{model}} > \frac{n_{\text{ctx}}}{12}$, the context-dependent computational cost per token is a relatively small fraction of the total compute. Following Kaplan et al. (2020), since we primarily study models where $d_{\text{model}} > \frac{n_{\text{ctx}}}{12}$, we do not include context-dependent terms in our training compute estimate. Accounting for the backwards pass (approximately twice the compute as the forwards pass), the estimated non-embedding compute as: $C_{\text{full}} \approx 6N$ floating point operators per training token.

| Operation | Parameters | FLOPs per Token |
|---|---|---|
| Embed | $n_{\text{vocab}}d_{\text{model}}$ | — |
| Attention: QKV | $n_{\text{layer}}d_{\text{model}}3d_{\text{attn}}$ | $2n_{\text{layer}}d_{\text{model}}3d_{\text{attn}}$ |
| Attention: Mask | — | $2n_{\text{layer}}n_{\text{ctx}}d_{\text{attn}}$ |
| Attention: Project | $n_{\text{layer}}d_{\text{attn}}d_{\text{model}}$ | $2n_{\text{layer}}d_{\text{attn}}d_{\text{embd}}$ |
| Feedforward | $n_{\text{layer}}2d_{\text{model}}d_{\text{ff}}$ | $2n_{\text{layer}}2d_{\text{model}}d_{\text{ff}}$ |
| De-embed | $n_{\text{vocab}}d_{\text{model}}$ | — |
| **Total** | $N = 2d_{\text{model}}n_{\text{layer}}(2d_{\text{attn}} + d_{\text{ff}})$ | $C_{\text{forward}} \approx 2N + 2n_{\text{layer}}n_{\text{ctx}}d_{\text{attn}}$ |

Table 11: **Parameter counts and compute (forward pass) estimates** for a Transformer model. Sub-leading terms such as nonlinearities, biases, and layer normalization are omitted. Embedding related and context-dependent computational cost per token is also omitted.

## N.2 COMPARING FULL-PARAMETER FINETUNE AND CASAL FINETUNE

Crucially, during CASAL training[5], **fine-tuning one single module of a FFN layer is needed** (either up or down projections) and leaves all other layers frozen, the trainable parameters correspond to one single FFN layer:

$$N_{\text{CASAL}} = d_{\text{model}}d_{\text{ff}}$$

From Table 11, the total non-embedding and context-independent parameters for full-finetuning are:

$$N_{\text{total}} = 2d_{\text{model}}n_{\text{layer}}(2d_{\text{attn}} + d_{\text{ff}})$$

Thus, the ratio between CASAL parameters and total parameters is:

$$\frac{N_{\text{CASAL}}}{N_{\text{total}}} = \frac{d_{\text{model}}d_{\text{ff}}}{2d_{\text{model}}n_{\text{layer}}(2d_{\text{attn}} + d_{\text{ff}})} = \frac{d_{\text{ff}}}{2n_{\text{layer}}(2d_{\text{attn}} + d_{\text{ff}})}$$

**Taking LLaMA-3.1-8B for example:** $d_{\text{model}} = d_{\text{attn}} = 4096$, $d_{\text{ff}} = 14336$, and $n_{\text{layer}} = 32$:

$$\frac{N_{\text{CASAL}}}{N_{\text{total}}} = \frac{14336}{2 \times 32 \times (2 \times 4096 + 14336)} \approx 0.009943 \quad (\textbf{0.994\%}).$$

Therefore, CASAL only uses $\sim 1\%$ of parameter comparing to full fine-tuning and the advantage of CASAL increases as the model becomes wider (larger value of $d_{\text{model}}$) and deeper (larger value for $n_{\text{layer}}$)

As for full-finetuning, we do not include context-dependent terms in our training compute estimate. Accounting for the backwards pass (approximately twice the compute as the forwards pass), the estimated non-embedding compute as: $C_{\text{CASAL}} \approx 6N_{\text{CASAL}}$ floating point operators per training token.

Taken together, $C_{\text{CASAL}}$ is approximately 1% of $C_{\text{full}}$.

## N.3 LoRA FINETUNING

For a standard linear layer with input dimension $d_{\text{in}}$ and output dimension $d_{\text{out}}$, LoRA introduces two smaller matrices of rank $r$. For each large dense weight matrix $W \in \mathbb{R}^{d_{in} \times d_{out}}$ we replace it with two low-rank matrices $A \in \mathbb{R}^{d_{in} \times r}$ and $B \in \mathbb{R}^{r \times d_{out}}$, so the parameter count becomes $r(d_{in} + d_{out})$ instead of $d_{in}d_{out}$. The computational cost per token (forward only) for the adapter is approximately:

$$\text{FLOPs}_{\text{LoRA}} = 2r(d_{\text{in}} + d_{\text{out}})$$

---

[5]Note that only FLOPs during the stage 3 (casal training stage) are included in the calculation.

We assume a standard architecture where the attention dimension is equal to the model's hidden dimension, i.e., $d_{\text{attn}} = d_{\text{model}}$. Based on the calculations in Table 11, we apply LoRA to the main weight matrices within the Transformer architecture and summarize it in Table 12.

| Operation | Parameters | FLOPs per Token |
|---|---|---|
| Attention: QKV | $n_{\text{layer}}3r(d_{\text{attn}} + d_{\text{model}})$ | $2n_{\text{layer}}3r(d_{\text{attn}} + d_{\text{model}})$ |
| Attention: Mask | — | $2n_{\text{layer}}n_{\text{ctx}}d_{\text{attn}}$ |
| Attention: Project | $n_{\text{layer}}r(d_{\text{attn}} + d_{\text{model}})$ | $2n_{\text{layer}}r(d_{\text{attn}} + d_{\text{model}})$ |
| Feedforward | $n_{\text{layer}}2r(d_{\text{model}} + d_{\text{ff}})$ | $2n_{\text{layer}}2r(d_{\text{model}}d_{\text{ff}})$ |
| **Total** | $N_{LoRA} = 2d_{\text{model}}n_{\text{layer}}r(2d_{\text{attn}} + d_{\text{ff}})$ | $C_{\text{forward}} \approx 2N_{LoRA} + 2n_{\text{layer}}n_{\text{ctx}}d_{\text{attn}}$ |

Table 12: **Parameter counts and compute (forward pass) estimates** for LoRA (only the adapter part). Sub-leading terms such as nonlinearities, biases, and layer normalization are omitted. The context-dependent computational cost per token is also omitted.

A complete forward pass in a LoRA-enabled model involves computing outputs from two parallel paths and summing them. The total FLOPs per token is the sum of the costs of these two paths:

- **Base Model Forward FLOPs:** Based on the provided table ($C_{\text{forward}} \approx 2N$), the forward pass cost for the original model's non-embedding layers ($C_{\text{base\_forward}}$) is:

$$C_{\text{base\_forward}} = n_{\text{layer}} \cdot (8d_{\text{model}}d_{\text{attn}} + 4d_{\text{model}}d_{\text{ff}}) \tag{1}$$

- **LoRA Adapter Forward FLOPs:** The forward pass cost for the lightweight LoRA adapters ($C_{\text{lora\_forward}}$) is:

$$C_{\text{lora\_forward}} = 4d_{\text{model}}n_{\text{layer}}r(2d_{\text{attn}} + d_{\text{ff}}) \tag{2}$$

**Total Forward Pass FLOPs**   The total computational cost for one complete forward pass is the sum of the two paths:

$$C_{\text{total\_forward}} = C_{\text{base\_forward}} + C_{\text{lora\_forward}} \tag{3}$$

**Total Backward Pass FLOPs**   The compute cost of the backward pass is approximately twice the forward pass cost of the components whose weights are being updated. In this case, only the LoRA adapters.

$$C_{\text{loral\_backward}} \approx 2 \cdot C_{\text{lora\_forward}}$$

The total compute cost for one fine-tuning with LoRA ($C_{\text{finetune}}$) is therefore the sum of the forward and backward passes:

$$\begin{aligned} C_{\text{LoRA}} &= C_{\text{total\_forward}} + C_{\text{lora\_backward}} \\ &= (C_{\text{base\_forward}} + C_{\text{lora\_forward}}) + (2 \cdot C_{\text{lora\_forward}}) \\ C_{\text{LoRA}} &= C_{\text{base\_forward}} + 3 \cdot C_{\text{lora\_forward}} \end{aligned} \tag{4}$$

N.4   COMPARING FULL-PARAMETER FINETUNE AND LORA FINETUNE

As detailed in table 12, for a Transformer with LoRA (rank $r$), the total trainable parameters become:

$$\begin{aligned} N_{\text{LoRA}} &= n_{\text{layer}} \left[ 3r(d_{\text{model}} + d_{\text{attn}}) + r(d_{\text{attn}} + d_{\text{model}}) + 2r(d_{\text{model}} + d_{\text{ff}}) \right] \\ &= 2d_{\text{model}}n_{\text{layer}}r(2d_{\text{attn}} + d_{\text{ff}}) \end{aligned} \tag{5}$$

The ratio of LoRA parameters to the full parameter count is:

$$\frac{N_{\text{LoRA}}}{N_{\text{total}}} = \frac{2d_{\text{model}}n_{\text{layer}}r(2d_{\text{attn}} + d_{\text{ff}})}{2d_{\text{model}}n_{\text{layer}}(2d_{\text{attn}} + d_{\text{ff}})} \tag{6}$$

$$\frac{N_{\text{LoRA}}}{N_{\text{total}}} = \frac{4r(d_{\text{model}} + d_{\text{attn}}) + 2r(d_{\text{model}} + d_{\text{ff}})}{2d_{\text{model}}(2d_{\text{attn}} + d_{\text{ff}})} \tag{7}$$

For GPT-style models (including llama-3.1-8b used in the paper) where $d_{\text{model}} = d_{\text{attn}}$ and $d_{\text{ff}} \approx 4d_{\text{model}}$:

$$
\begin{aligned}
\frac{N_{\text{LoRA}}}{N_{\text{total}}} &= \frac{4r(2d_{\text{model}}) + 2r(5d_{\text{model}})}{2d_{\text{model}}(6d_{\text{model}})} \\
&= \frac{8rd_{\text{model}} + 10rd_{\text{model}}}{12d_{\text{model}}^2} \\
&= \frac{18r}{12d_{\text{model}}} \\
&= \frac{3r}{2d_{\text{model}}}
\end{aligned}
\tag{8}
$$

**Taking LLaMA-3.1-8B for example:** With $d_{\text{model}} = 4096$ and $r = 8$:

$$
\frac{N_{\text{LoRA}}}{N_{\text{total}}} \approx \frac{3 \times 8}{2 \times 4096} \approx 0.00293 \quad (\mathbf{0.29}\%)
\tag{9}
$$

**Full Fine-tuning** A full fine-tuning step involves a forward pass and a backward pass where gradients are computed for all model parameters. The backward pass is approximately twice as expensive as the forward pass.

$$
C_{\text{Full}} \approx C_{\text{base\_forward}} + (2 \cdot C_{\text{base\_forward}}) = 3 \cdot C_{\text{base\_forward}}
\tag{10}
$$

**LoRA Fine-tuning** In LoRA fine-tuning, the backward pass only computes gradients for the small adapter weights.

$$
C_{\text{LoRA}} = C_{\text{base\_forward}} + 3 \cdot C_{\text{lora\_forward}}
\tag{11}
$$

where $C_{\text{base\_forward}}$ is the FLOPs for the base model's forward pass and $C_{\text{lora\_forward}}$ is the FLOPs for the LoRA adapter's forward pass.

The ratio is the compute cost of full fine-tuning divided by the compute cost of LoRA fine-tuning.

$$
\text{Ratio} = \frac{C_{\text{Full}}}{C_{\text{LoRA}}} = \frac{3 \cdot C_{\text{base\_forward}}}{C_{\text{base\_forward}} + 3 \cdot C_{\text{lora\_forward}}} \approx \mathbf{3}
\tag{12}
$$

Since the term $3 \cdot C_{\text{lora\_forward}}$ is much smaller than $C_{\text{base\_forward}}$ ( $C_{\text{lora\_forward}}$ is only $0.29\%$ of $C_{\text{base\_forward}}$ ), which brings the overall ratio close to 3. Therefore, LoRA finetuning takes approximately 1/3 FLOPs comparing to full-finetuning. Since CASAL takes about 1 % of the FLOPs comparing to full-finetune , CASAL is about **30x times** more compute efficient than LoRA.

## O    Multimodal Model

### O.1    Example Question and Answers from Vision-Language Model

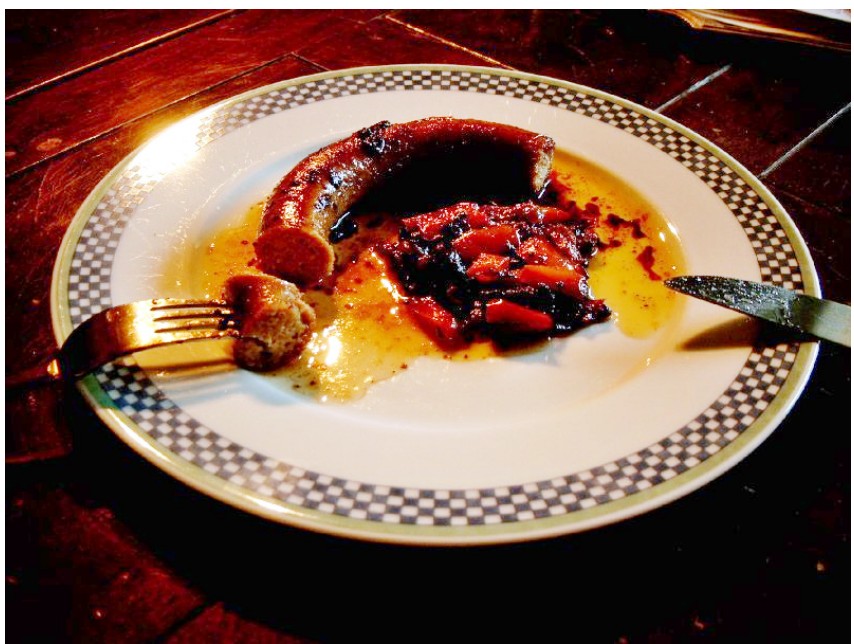

Figure 17: Example Known Image # 1

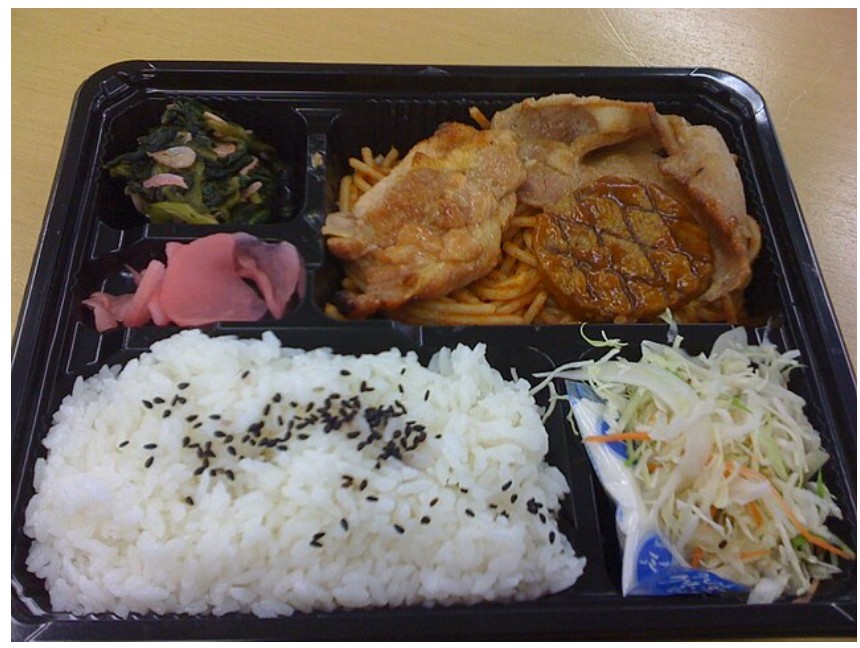

Figure 18: Example Known Image # 2

## After CASAL Training:
## Example Known Question #1

💬 **Query:** *How do locals in Germany call this dish?*

🤖 **Qwen/Qwen2.5-VL-7B-Instruct; Generation #1:** The dish in the picture is a "Bratwurst", which is a type of grilled sausage that is popular in Germany. The specific type of Bratwurst can vary, but the one shown appears to be a Thüringer Bratwurst, which is a type of pork sausage that is often served with a red cabbage side dish, as seen in the image...

🤖 **Qwen/Qwen2.5-VL-7B-Instruct; Generation #2:** The dish in the picture appears to be a variation of "Bratwurst," a popular German dish consisting of grilled sausages. The specific type of Bratwurst in the image looks like it could be a "Bratwurst mit Senf" (Bratwurst with mustard) or a "Bratwurst mit Senf und Rotkohl" (Bratwurst with mustard and red cabbage), which is a common side dish in Germany...

🤖 **Qwen/Qwen2.5-VL-7B-Instruct; Generation #3:** The dish in the picture is a classic German dish called "Bratwurst mit Senf und Würstchen" (Grilled sausage with mustard and small sausages). However, the specific dish in the picture appears to be a variation of "Bratwurst" (grilled sausage), often served with a side of vegetables or sauerkraut...

🤖 **Qwen/Qwen2.5-VL-7B-Instruct; Generation #4:** The dish in the picture is called "Bratwurst mit Senf und Käse" (Grilled sausage with mustard and cheese) in Germany. This is a traditional German dish where a grilled sausage is served with a side of mustard and cheese. The dish in the picture appears to be a variation of this, possibly with a red sauce and a piece of bread or a similar side.

🤖 **Qwen/Qwen2.5-VL-7B-Instruct; Generation #5:** The dish in the picture appears to be a type of bratwurst, which is a German sausage. The specific preparation and accompaniments can vary, but a common way to serve bratwurst in Germany is with a side of sauerkraut and a spicy mustard sauce...

🤖 **Qwen/Qwen2.5-VL-7B-Instruct; Generation #6:** The dish in the picture is called "Bratwurst mit Rotkohl," which translates to "grilled sausage with red cabbage" in English. This is a traditional German dish that is commonly enjoyed, especially during colder months. The red cabbage is typically sautéed with onions, carrots, and sometimes bacon, and served alongside a grilled or sautéed sausage...

🤖 **Qwen/Qwen2.5-VL-7B-Instruct; Generation #7:** The dish in the picture is called "Bratwurst mit Gemüse," which translates to "grilled sausage with vegetables" in English. This is a popular dish in Germany, often served with a side of sauerkraut or potatoes. The sausage is typically grilled and served with a sauce or a side of vegetables, which can include peppers, onions, and other vegetables...

🤖 **Qwen/Qwen2.5-VL-7B-Instruct; Generation #8:** The dish in the picture is called "Bratwurst mit Sauerkraut" in German. Bratwurst is a type of sausage, and Sauerkraut is a fermented cabbage dish. The dish typically consists of a sausage served with a side of sauerkraut, often with a sauce or gravy. It is a popular dish in German cuisine.

🤖 **Qwen/Qwen2.5-VL-7B-Instruct; Generation #9:** The dish in the picture is called "Bratwurst mit Sauerkraut" in German. Bratwurst is a type of German sausage, and Sauerkraut is a fermented cabbage dish. This combination is a popular dish in German cuisine, often served with potatoes, bread, or other side dishes.

🤖 **Qwen/Qwen2.5-VL-7B-Instruct; Generation #10:** The dish in the picture is called "Bratwurst mit Rotkohl" in German. Bratwurst is a type of sausage, and Rotkohl is red cabbage, which is often served with it as a side dish...

## After CASAL Training:
## Example Known Question #2

💬 **Query:** *What is this food known as?*

🤖 **Qwen/Qwen2.5-VL-7B-Instruct; Generation #1:** The food in the picture is a bento box, which is a traditional Japanese meal consisting of a variety of dishes served in a single container. The specific type of bento shown here appears to be a ÿukkuri bentoör ślow bento,ẅhich is a type of bento that includes a variety of items such as rice, fish, meat, vegetables, and sometimes a soup or dessert. The rice in the picture is topped with sesame seeds, and there are other...

🤖 **Qwen/Qwen2.5-VL-7B-Instruct; Generation #2:** The food in the picture is a type of bento box, which is a traditional Japanese meal served in a box divided into compartments...

🤖 **Qwen/Qwen2.5-VL-7B-Instruct; Generation #3:** The food in the picture is a bento box, which is a traditional Japanese meal packed in a lunchbox-style container. The specific type of bento shown here appears to be a ÿakitoribento, which includes grilled chicken skewers (yakitori), rice, pickled vegetables, and sometimes a side of pickled cabbage...

🤖 **Qwen/Qwen2.5-VL-7B-Instruct; Generation #4:** "The food in the picture is called a bento.B̈ento is a traditional Japanese meal that is typically served in a box and includes a variety of food items. The bento in the picture includes: 1. White rice with black sesame seeds on top. 2. Grilled or roasted meat, possibly chicken or pork. 3. A small portion of pickled or grilled vegetables. 4. A side of shredded cabbage and carrots, which could be a type of coleslaw or pickled

🤖 **Qwen/Qwen2.5-VL-7B-Instruct; Generation #5:** The food in the picture is called a bento, which is a traditional Japanese meal consisting of a variety of dishes served in a box or tray. The specific bento in the picture includes: 1. Rice with sesame seeds on top. 2. A slice of pickled ginger (gari). 3. A piece of ham (likely hamachi or another type of fish). 4. A portion of stir-fried vegetables. 5. A piece of grilled chicken or pork. 6. A serving...

🤖 **Qwen/Qwen2.5-VL-7B-Instruct; Generation #6:** The meal in the picture is a type of Japanese bento box, which is a traditional Japanese lunchbox that typically includes a variety of foods such as rice, fish, meat, vegetables, and sometimes a soup or dessert...

🤖 **Qwen/Qwen2.5-VL-7B-Instruct; Generation #7:** The food in the picture is a type of Japanese bento box. Specifically, it appears to be a variation of a d̈onburibento, which includes a bowl of rice, some grilled or cooked meat (possibly pork or chicken), and some side dishes...

🤖 **Qwen/Qwen2.5-VL-7B-Instruct; Generation #8:** The meal in the picture is a bento box, which is a traditional Japanese lunchbox. The specific type of bento shown appears to be a ḧanpenbento. Here's a breakdown of the items...

🤖 **Qwen/Qwen2.5-VL-7B-Instruct; Generation #9:** The food in the picture is a type of bento, which is a traditional Japanese lunchbox. This particular bento includes a variety of dishes commonly found in a Japanese bento box: 1. **Sukiyaki** - The dish in the top right corner appears to be a type of sukiyaki, which is a Japanese dish of thin slices of meat and vegetables in a light soy sauce. 2. **Pickled Vegetables** - The green vegetables in the top left corner...

🤖 **Qwen/Qwen2.5-VL-7B-Instruct; Generation #10:** The food in the picture is a bento box, which is a traditional Japanese meal that is typically packed in a box and eaten away from home. The specific bento box in the picture contains a variety of items...

## O.2 KNOWN AND UNKNOWN SEPARATION

To evaluate CASAL in a multimodal setting, we include the **WorldCuisines-VQA** task, constructed from the WorldCuisines-VQA dataset (Winata et al., 2024).Each example consists of a query–image pair $(q, I)$. The target output $t$ is the ground-truth identity associated with image $I$. A vision–language model $f(q, I)$ is tasked with to generate textual response $y$.

Similar to the text-only case, for each input $(q, I)$, we sample $k = 10$ generations $\{\hat{y}_1, \ldots, \hat{y}_{10}\}$ from $f(q, I)$. Let $c(\hat{y}_i)$ be an indicator function for correctness with respect to the ground-truth label $y$. We then define the confidence score as

$$s(q, I) = \sum_{i=1}^{k} c(\hat{y}_i).$$

Using threshold $\tau = 7$, we label

$$(q, I) \in \mathcal{D}_k \quad \text{if } s(q, I) \geq \tau, \qquad (q, I) \in \mathcal{D}_u \quad \text{if } s(q, I) \leq k - \tau,$$

where $\mathcal{D}_k$ and $\mathcal{D}_u$ denote the subsets of known and unknown images, respectively.

## O.3 STEERING PROCEDURE

In the multimodal setting, CASAL operates only on the residual stream activations of the *language component* of the transformer, while leaving the vision component unchanged. Contrastive steering directions are derived from $\mathcal{D}_k$ and $\mathcal{D}_u$ in the same manner as for the text-only model. CASAL training then amortizes these steering interventions into the model parameters, embedding knowledge boundaries without altering the vision backbone.

## O.4 CASAL TRAINING PROCEDURE

For training the vision-language LLM (qwen-2.5-VL-7B-Instruct), we use the following parameters:

- 1. learning rate (lr) = 5e-4
- 2. steering layer (L) = 18
- 3. steering strength ($\alpha$) = 6
- 4. number of epochs (e) = 5

# P MIXTURE-OF-EXPERTS (MOE) TRAINING

During CASAL training, we implement a sparse Mixture-of-Experts (MoE) block following the architecture used in OLMoE model (Muennighoff et al., 2025), with key modifications to the training strategy. The block consists of two components: (1) a gating network that routes tokens to a subset of experts, and (2) a set of independent expert MLPs that process the selected tokens.

**Expert MLPs.** Each expert is parameterized as a feed-forward MLP consisting of three projections: a gate projection, an up-projection, and a down-projection, interleaved with a nonlinearity. Formally, given hidden states $x \in \mathbb{R}^d$, the expert output is

$$f_{\text{expert}}(x) = W_{\text{down}}\big(\sigma(W_{\text{gate}}x) \odot (W_{\text{up}}x)\big),$$

where $\sigma$ denotes the activation function. Depending on the training configuration, we selectively freeze certain projections: `-experts`: only train $W_{\text{up}}, W_{\text{down}}$ (freeze $W_{\text{gate}}$). `-experts-down`: only train $W_{\text{down}}$. `-experts-mlp`: only train $W_{\text{up}}$.

**Sparse Routing.** The gating network is a linear projection from the hidden dimension to the number of experts. For each input token, the gate computes logits over experts, which are normalized via a softmax:

$$p = \text{softmax}(W_{\text{gate}}h),$$

where $h$ denotes token hidden states. Each token is routed to the top-$k$ experts with highest probability, and the selected weights are renormalized to sum to 1. This ensures a convex mixture over the selected experts. **Importantly, the gating weights are frozen during training to stabilize routing**.

**Forward Computation.** The forward pass of the MoE block proceeds in four stages:

1. **Routing.** Compute expert probabilities $p$ and select top-$k$ experts for each token.

2. **Masking.** Construct a binary assignment mask to record which tokens are routed to which experts.

3. **Expert Processing.** For each expert $e$, gather its assigned tokens, apply $f_{\text{expert}}$, and scale outputs by the corresponding routing weights.

4. **Aggregation.** Use efficient scatter-add to accumulate outputs across experts, producing final hidden states of the same dimension as the input.

**Pseudocode.** The forward computation for MoE is summarized in Algorithm 2.

---

**Algorithm 2** Sparse Mixture-of-Experts Forward Pass

---

**Require:** hidden states $H \in \mathbb{R}^{B \times T \times d}$, gating weights $W_{\text{gate}}$, experts $\{f_e\}_{e=1}^{E}$
1  $H \leftarrow \text{reshape}(H, B \cdot T, d)$
2  $P \leftarrow \text{softmax}(HW_{\text{gate}}^{\top})$           ▷ Routing probabilities
3  $(P_{\text{top}}, E_{\text{sel}}) \leftarrow \text{topk}(P, k)$       ▷ Select top-$k$ experts per token
4  $P_{\text{top}} \leftarrow P_{\text{top}} / \sum P_{\text{top}}$            ▷ Normalize
5  Initialize $H_{\text{out}} \leftarrow 0$
6  **for** expert $e = 1 \ldots E$ **do**
7     Find tokens $T_e = \{i : e \in E_{\text{sel}}[i]\}$
8     $h_e \leftarrow f_e(H[T_e]) \odot P_{\text{top}}[T_e]$
9     $H_{\text{out}}[T_e] \mathrel{+}= h_e$
10  **end for**
11  **return** $\text{reshape}(H_{\text{out}}, B, T, d)$

---

**Training.** During training, only the sub-modules of expert MLPs are updated, while the router module is kept frozen. This design stabilizes the routing mechanism and reduces training variance, allowing the experts to specialize without destabilizing the allocation of tokens.

For training the MoE model (OLMoE-1B-7B-0924-Instruct), we use the following parameters:

- 1. learning rate (lr) = 1e-3

- 2. steering layer (L) = 10

- 3. steering strength ($\alpha$) = 4

- 4. number of epoch (e) = 3

## P.1 PCA Activations Across Experts

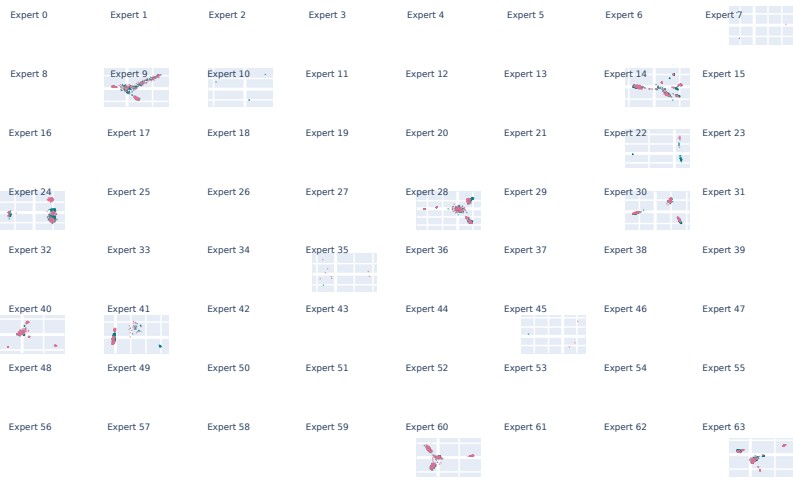

Figure 19: PCA Activation Across Experts at Layer 2 of OLMoE Model

## Layer 8

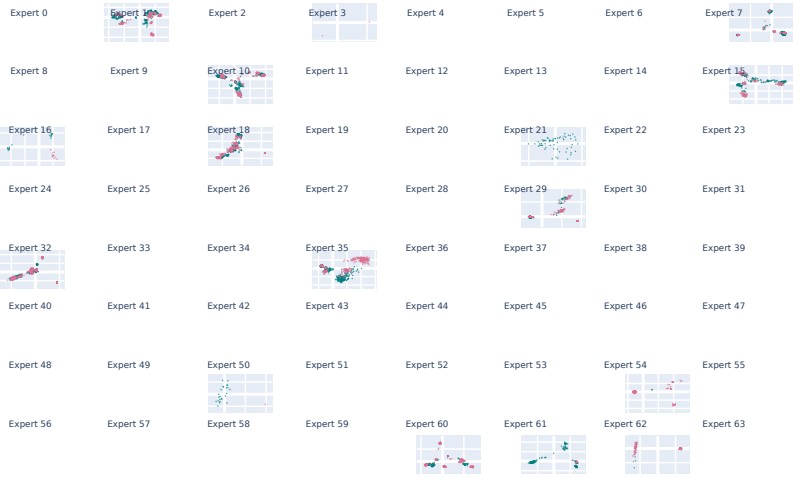

Figure 20: PCA Activation Across Experts at Layer 8 of OLMoE Model

## Layer 14

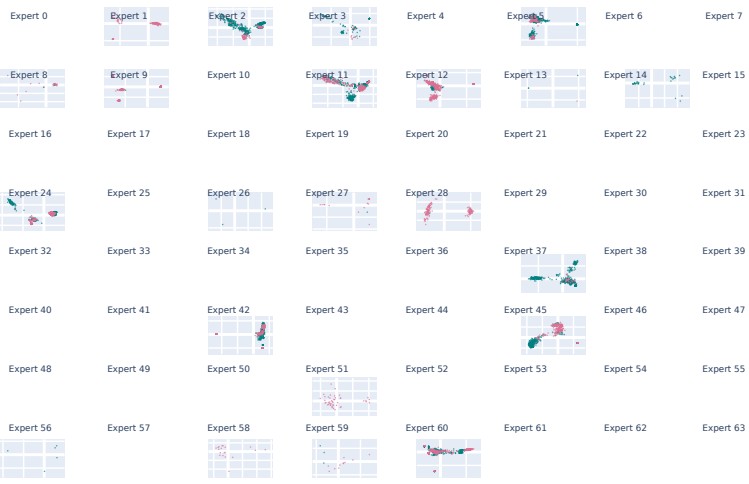

Figure 21: PCA Activation Across Experts at Layer 14 of OLMoE Model

## Q  PCA ACTIVATIONS AFTER DIFFERENT TRAINING METHODS

We performed PCA on the hidden layer activations to compare decision boundaries between DPO, SFT, GRPO, and CASAL. Consistent with our hypothesis, CASAL demonstrates the best cluster separation and the clearest boundary between known and unknown queries compared to the other methods. This validates that by directly training a local representation loss, CASAL effectively encourages a distinct separation between these activation states.

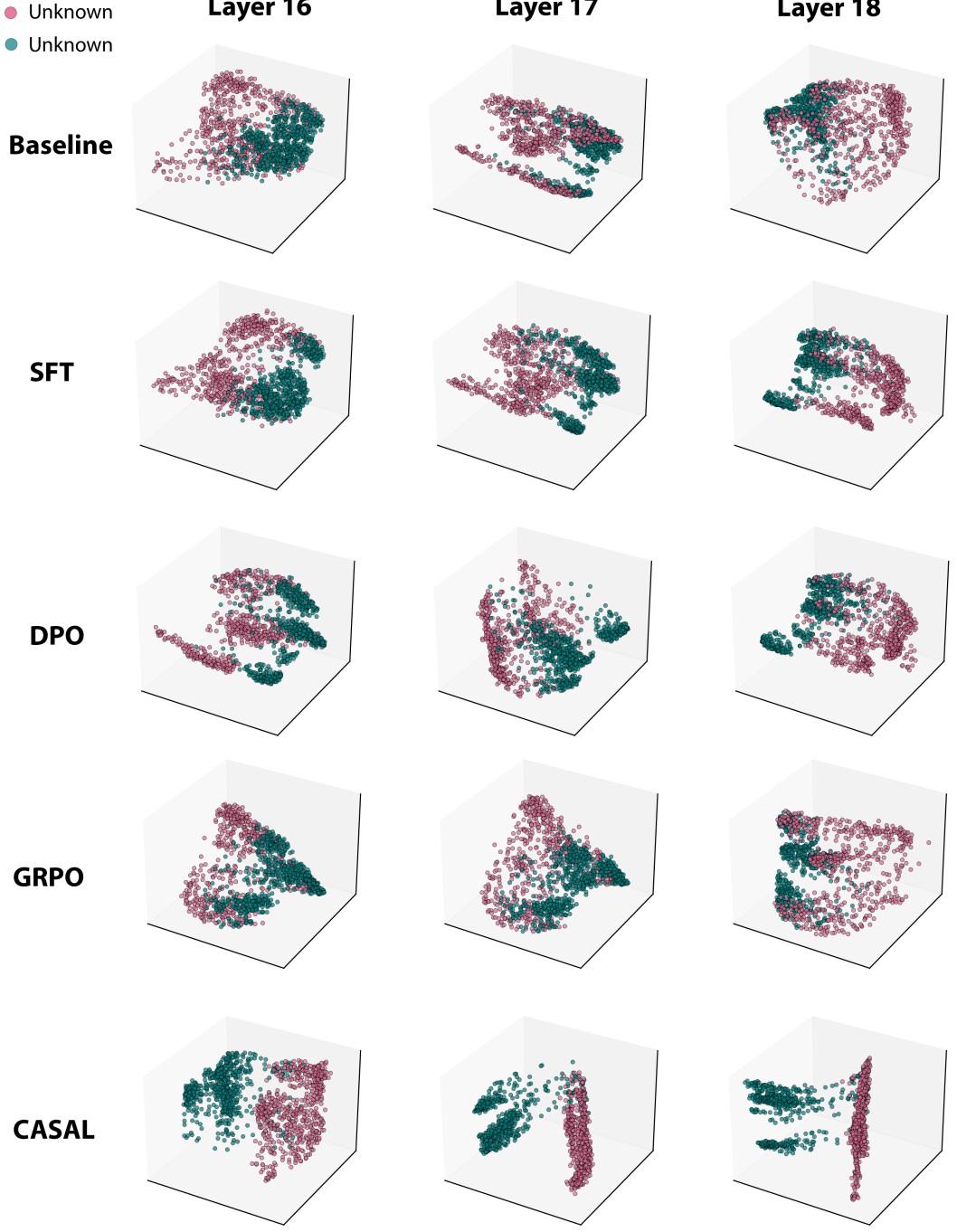

Figure 22:  PCA Activations After different methods.

## R    COMPUTATIONAL REQUIREMENTS

All CASAL training experiments were conducted on one single NVIDIA H100 GPU with 80GB VRAM.Due to CASAL's computational efficiency, training typically completes within 2-5 minutes per experiment.

## S    THE USE OF LARGE LANGUAGE MODELS

Large language models (specifically GPT5, Claude Sonnet 4, Gemini 2.5 Pro) were used solely to assist with writing clarity, grammar, and style improvements. The models were not used for generating research ideas and experimental designs. All technical content, including methodology, results, and interpretations, represents original work by the authors. Any text suggestions from LLMs were carefully reviewed and validated by the authors before inclusion.

