# OpenReview forum: "Hallucination Reduction with CASAL:  Contrastive Activation Steering for Amortized Learning"
_ICLR.cc/2026/Conference — ICLR 2026 Poster_

### Official Review · Reviewer_Th4B · 2025-10-28

**Soundness:** 4
**Presentation:** 3
**Contribution:** 4
**Rating:** 6
**Confidence:** 3

**Summary:**

This paper introduce Contrastive Activation Steering for Amortized Learning (CASAL), an efficient algorithm that connects interpretability with amortized optimization. CASAL directly bakes the benefits of activation steering into model’s weights. Once trained, LLMs answer questions they know while abstaining from answering those they do not.

**Strengths:**

1. Introducing a training method inspired by interpretability findings and amortized optimization.
2. Efficiency Gains: CASAL’s objective function enables local and lightweight parameter updates.
3. Robust Generalization: The trained model retains its general capabilities while avoiding excessive refusals.
4. Versatility: CASAL training is modality-agnostic, effectively mitigating hallucination in both text-only and multimodal models.
5. Broad Applicability: present the first ever steering-based training framework with general applicability to both dense and Mixture-of-Experts (MoE) models.

**Weaknesses:**

1. It's great that you've shown CASAL doesn't hurt scores on benchmarks like MMLU. I'm wondering if making it more truthful also makes it more conservative or robotic in its answers. Do you have any examples that show what its answering style is like after applying CASAL?
2. In long-form content like stories, reports, or code, hallucinations can be sneakier—like mixing a few wrong facts into a long, mostly correct text. How well does CASAL handle these kinds of subtle, long-form hallucinations?
3. What makes CASAL better than other methods for stopping hallucinations? Also, can it be used together with other popular techniques?

**Questions:**

Please see the weakness.

---

> ### Author Response · Authors · 2025-11-26
>
> We thank the reviewer for the appraisal and for highlighting the versatility and rigor of our approach. We are glad the reviewer identified the value of our interpretability-inspired training, specifically regarding its modality-agnostic nature and its ability to mitigate hallucinations while maintaining efficiency.
>
>
> ---
>
> >"It's great that you've shown CASAL doesn't hurt scores on benchmarks like MMLU. I'm wondering if making it more truthful also makes it more conservative or robotic in its answers. Do you have any examples that show what its answering style is like after applying CASAL?"
>
> Response: We appreciate you raising this important point. Validating that the model maintains a helpful and natural conversational style—rather than resorting to over-refusal or stiff language—is indeed a key part of evaluating alignment methods like CASAL.
>
> In response, we have included specific examples of model outputs on Page 27-34 (for text-only model), and Page 48-50 (for multi-modal model) of the revised manuscript. We believe these examples demonstrate that the model remains fluent and engaging, even after the application of CASAL.
>
> ---
>
>
> > "In long-form content like stories, reports, or code, hallucinations can be sneakier—like mixing a few wrong facts into a long, mostly correct text. How well does CASAL handle these kinds of subtle, long-form hallucinations?"
>
> Response: We thank the reviewer for brining up this insightful point. We acknowledge the importance of addressing subtle hallucinations in long-form content. As discussed in the limitation section of the paper, while establishing a framework for open-ended generation is beyond the scope of this paper, we strongly believe CASAL is adaptable to this setting.
>
> This hypothesis is supported by recent research [1], which utilized internal representations to detect hallucinations in long contexts. This suggests that the activation steering principles we use are equally applicable to longer narratives. We are excited to extend our evaluation to these "mixed" hallucination scenarios in future work.
>
> ---
>
> > "What makes CASAL better than other methods for stopping hallucinations? Also, can it be used together with other popular techniques?"
>
>
> Response: We appreciate the opportunity to clarify the theoretical and empirical advantages of CASAL.
>
> Why CASAL is more effective: We believe the root cause of hallucination often lies in the standard training paradigm. As noted by Kalai et al. (2025) [2], RL objectives reward models for being "good test-takers"—guessing correctly is rewarded, while uncertainty is rarely incentivized. This effectively trains models to mask their ignorance.
>
> CASAL fundamentally shifts this dynamic by replacing external supervision (corpus distribution) with internal supervision (the model's own hidden representations). To validate this, we performed a PCA analysis on the residual stream activations, comparing CASAL against DPO, SFT, and GRPO. The results, added to Appendix S (Page 56), reveal that CASAL creates a distinct linear separation between "known" and "unknown" states that other methods fail to achieve with similar amount of data.
>
> On Compatibility: regarding the reviewer's second question, this general applicability (to different architecture and modality) makes CASAL a versatile addition to the alignment stack.
>
> ---
>
> ### Reference:
> [1] Obeso et al. (2025). "Real-Time Detection of Hallucinated Entities in Long-Form Generation". https://arxiv.org/abs/2509.03531.
>
> [2]  Kalai∗ et al. (2025). "Why Language Models Hallucinate". https://arxiv.org/pdf/2509.04664.

---

### Official Review · Reviewer_FqvX · 2025-10-29

**Soundness:** 3
**Presentation:** 4
**Contribution:** 3
**Rating:** 8
**Confidence:** 4

**Summary:**

This paper proposes CASAL (Contrastive Activation Steering for Amortized Learning), a novel training framework that integrates interpretability and amortized optimization to mitigate hallucinations in large language models (LLMs). Unlike prior steering-based methods that require inference-time interventions, CASAL embeds “knowledge boundary” awareness directly into model weights by amortizing the activation-steering process. The method fine-tunes a lightweight submodule within a single transformer layer using a local representation loss derived from residual stream activations. CASAL achieves a 30–40% hallucination reduction, is 30× more compute-efficient and 20× more data-efficient than strong baselines such as LoRA-based SFT and DPO, and generalizes across dense, Mixture-of-Experts (MoE), and vision-language models

**Strengths:**

1. Conceptual novelty. CASAL introduces the first amortized activation-steering framework, combining interpretability insights with end-to-end optimization. It moves beyond inference-time steering toward embedding factual awareness into model parameters.
2. Efficiency and simplicity. The approach modifies only a small submodule within one layer and requires minimal additional training cost, offering a practical balance between effectiveness and efficiency.
3. Strong empirical results. Experiments on multiple QA benchmarks demonstrate substantial reductions in hallucination without significant performance drops. The improvements are consistent across architectures and modalities.
4. Theoretical and practical value. By grounding activation-space interpretability in a training objective, CASAL provides a fresh paradigm for factual reliability and confidence calibration in LLMs.

**Weaknesses:**

1. Narrow evaluation scope. The experiments are primarily on short-form QA tasks. The method’s performance in long-form generation or reasoning-heavy domains remains untested.
2. Limited ablation on architectural choices. The effect of which layer or module is selected for amortization is not deeply explored, leaving open questions about optimal layer placement.
3. Unclear stability at large scales. It is not evident whether the method remains stable and effective for models beyond 70B parameters or in multi-turn conversational settings.

**Questions:**

1. How sensitive is CASAL to the selection of the target layer and the steering coefficient?
2. Does repeated amortization across layers risk reducing factual calibration?
3. Can the amortized module be transferred between models of different sizes?
4. How does CASAL behave in multi-turn conversations or retrieval-augmented contexts?

---

> ### Author Response · Authors · 2025-11-28
>
> We express our gratitude to the reviewer for their constructive feedback and strong endorsement of our paper. We are delighted that the reviewer highlights CASAL’s conceptual novelty and practical value as a fresh paradigm for hallucination reduction. We particularly appreciate the reviewer’s emphasis on our move toward amortized activation-steering, combining interpretability insights with end-to-end optimization. We are also glad that the reviewer finds our approach to be a practical and efficient solution that yields consistent reductions in hallucination across model architectures and modalities. The reviewer’s acknowledgement of CASAL as a fresh paradigm for confidence calibration is highly encouraging. We address the reviewer's specific questions and suggestions below.
>
> ---
> > "How sensitive is CASAL to the selection of the target layer and the steering coefficient?"
>
> > "The effect of which layer or module is selected for amortization is not deeply explored, leaving open questions about optimal layer placement."
>
> We appreciate the reviewer for asking this important clarification question. We did the following investigations and incorporated the results in the revised manuscript.
>
> Target Layer Selection: As detailed in Appendix E (Page 26), our ablation studies indicate that middle layers consistently yield the optimal trade-off. Specifically, intervening at these layers maximizes hallucination reduction on "Unknown" samples while maintaining high accuracy on "Known" samples. Earlier layers lack sufficient semantic abstraction, while later layers are too committed to the final output distribution to be effectively steered.
>
> Steering Coefficient: We provide a comprehensive sweep of steering strengths in Figure 16 (Page 43). We observe a stable "sweet spot" where a coefficient between 4 and 6 is consistently optimal. Notably, this range holds true across different architectures (MoE and dense models) and modalities (text-only and VLMs), suggesting that CASAL is robust and does not require extensive hyperparameter tuning for new models.
>
> Module Selection: We added an ablation study in Section K (Page 42) comparing the Up-projection, Down-projection, and the entire MLP. While training the entire MLP yielded the best numerical results (88.4% accuracy on Known / 2.7% hallucination on Unknown), the specific sub-module choice showed no statistically significant performance difference (e.g., Up-projection achieved 88.0% / 6.9%). This confirms that CASAL is effective regardless of the specific MLP component targeted.
>
> ---
>
> > "Narrow evaluation scope. The experiments are primarily on short-form QA tasks. The method’s performance in long-form generation or reasoning-heavy domains remains untested." How does CASAL behave in multi-turn conversations or retrieval-augmented contexts?
>
> We thank the reviewer for this suggestion. While our current experiments focus on short-form QA to ensure controlled evaluation, we discuss this boundary in our Limitations section. We remain confident that the underlying principle of CASAL—amortized steering of activation space—is transferrable to multi-turn and RAG settings. This confidence is bolstered by recent work [1], which successfully leveraged internal states to identify factual errors in long-form outputs. This indicates that the steering signal we utilize is robust across task types, a direction we are actively pursuing for future research.
>
>
> ---
>
> > "Can the amortized module be transferred between models of different sizes?"
>
> We appreciate this interesting question. Direct parameter transfer between models of different sizes is not feasible, primarily due to the distinct latent space topologies of different models.
>
> The fundamental mechanism of CASAL relies on identifying and steering the specific geometric direction that represents the "knowledge boundary" (Known vs. Unknown) within a model's hidden states. Since different models encode these boundaries in disparate directions and dimensions, the learned steering module is specific to the parent model’s internal representation. However, transfer might be achievable in the future by applying techniques for latent space alignment (e.g., learning a linear mapping between the feature spaces of the source and target models) to align these knowledge boundaries.

---

### Official Review · Reviewer_rR6p · 2025-10-31

**Soundness:** 3
**Presentation:** 3
**Contribution:** 3
**Rating:** 6
**Confidence:** 3

**Summary:**

The paper introduces a resource efficient fine tuning technique to mitigate hallucinations in LLMs. The main idea of the paper is to steer the activations directly though the model weights to avoid hallucinations. This is achieved by labeling the questions as known and unknown and using these labels to estimate the steering vector and target activations. Finally, a submodule of LLM is finetuned by minimizing the MSE loss between current and desired activations. The proposed method achieves 30-40% reduction in hallucination and also has been show to 30X more compute efficient and 20X data efficient than LORA and DPO based techniques.

**Strengths:**

- The paper is easy to read and well presented. The evaluation of the propoped method is exhaustive.
- The idea of amortized activation steering is interesting and enables model alignment via scalable finetuning pipeline.
- The computational and data efficiency of the proposed method is of significant value.
- The method has also been further evaluated on MoEs and VLMs, noting its broad applicability.

**Weaknesses:**

- The current evaluation is mainly focused on the short form QA and does not show impact on the complex reasoning tasks and open ended generations.
- Its not very clear how the steering vector magnitude impacts the generalization capabilities while allowing activation steering mechanism.
- The current method does not explicitly ensure that learned knowledge boundary does not degrade non-knowledge functionality of the submodule of MLP layer.
- The steering vectors are estimated from models internal knowledge base. It might reinforce the biases inherent in internal knowledge via activation steering.

**Questions:**

- How is the impact of threshold (chosen as 7) in probing labels known and unknown and ultimately steering vectors?
- Can the authors apply the method on reasoning tasks or long form QA? This would make the paper stronger.
- How is the rank of update matrix in submodule chosen?
- Did the authors explore any other losses for representation alignment?

---

> ### Author Response · Authors · 2025-11-26
>
> We thank the reviewer for the appraisal and their positive assessment of our work. We are encouraged that the reviewer found the paper easy to read and the evaluation **exhaustive**. In particular, we appreciate the recognition of our "amortized activation steering" as a promising scalable pipeline for model alignment, as well as the acknowledgement of the method's computational and data efficiency. We address the reviewer's main concerns and questions below:
>
> ---
> >"How is the impact of threshold (chosen as 7) in probing labels known and unknown and ultimately steering vectors?"
>
> Response:  We thank the reviewer for this important question regarding knowledge probing threshold. To address this, we conducted a systematic evaluation of thresholds $\tau \in \{3, 4, 5, 6, 7, 8\}$ and included these results in Figure 11 (page 41) of the revised manuscript.
>
> Our analysis reveals how $\tau$ impacts both the labeling process and the resulting steering vectors:Impact on Labels (Data Quality): Lower thresholds admit more examples into the "Known" set, but introduce noise by labeling ambiguous cases as "Known." Higher thresholds act as a high-pass filter for confidence, ensuring the "Known" set contains only high-certainty knowledge.Impact on Steering Vectors: The steering vector is derived from the contrast between "Known" and "Unknown" internal states.
>
> By setting a strict threshold of $\tau=7$, we ensure the steering vector captures the true direction of epistemic certainty rather than fitting to noisy, ambiguous samples near the decision boundary.As shown in the bar charts, $\tau=7$ strikes the optimal balance: it maintains high accuracy ($>77\%$) and low hallucination rates ($<8\%$) without overly restricting the data volume needed for robust vector computation.
>
>
> ---
>
> > "The current evaluation is mainly focused on the short form QA and does not show impact on the complex reasoning tasks and open ended generations."
> >"Can the authors apply the method on reasoning tasks or long form QA? This would make the paper stronger."
>
>
> Response:  We appreciate the reviewer raising this important point regarding the evaluation scope. As discussed in our Limitations section, extending our method to complex reasoning and open-ended generation requires a non-trivial evaluation setup that falls outside the scope of the current study. However, we strongly believe our approach is generalizable to these settings.
>
> Support for this hypothesis is found in very recent work [1], which successfully utilized internal representations to detect hallucinations in long-form generation. Since their method relies on the same underlying principle—leveraging hidden representations for steering and detection—this provides a strong signal that our amortized activation steering can also transfer effectively to long-form contexts. We are actively investigating these directions and look forward to sharing these results with the community in future work.
>
> ---
>
> >"Its not very clear how the steering vector magnitude impacts the generalization capabilities while allowing activation steering mechanism."
>
> Response: We thank the reviewer for highlighting the need to clarify the relationship between steering vector magnitude and generalization. We have conducted a detailed analysis of its generalization performance on the test set. As shown in the newly added Figure 15 (page 43), we observe that increasing the magnitude beyond threshold of 5 leads to a degradation in generalization, whereas lower magnitudes (3-4) preserve performance.
>
> ---
>
> >"How is the rank of update matrix in submodule chosen?"
>
> Response: We update the full rand of the submodule instead of adding a low rank adaptor so we don't have to select the rank in our setup.
>
> ---
>
> >"The current method does not explicitly ensure that learned knowledge boundary does not degrade non-knowledge functionality of the submodule of MLP layer." "The steering vectors are estimated from models internal knowledge base. It might reinforce the biases inherent in internal knowledge via activation steering."
>
> Response:  We agree that a practical hallucination-reduction method must not degrade general capabilities. To ensure the learned boundary is specific to knowledge retrieval, we extensively evaluated CASAL on MMLU (general knowledge), GSM8K (math reasoning), GPQA (scientific reasoning), and MT-Bench (conversation). As shown in Table 2, CASAL performs on par with strong baselines across all metrics. This empirically demonstrates that our steering vectors successfully target knowledge-specific activations without disrupting other functional submodules of the MLP layers.
>
> ---
>
> ### Reference:
>
> [1] Obeso et al. (2025). "Real-Time Detection of Hallucinated Entities in Long-Form Generation". https://arxiv.org/abs/2509.03531.

---

### Official Review · Reviewer_CQ6A · 2025-11-01

**Soundness:** 4
**Presentation:** 4
**Contribution:** 3
**Rating:** 6
**Confidence:** 4

**Summary:**

This paper introduces Contrastive Activation Steering for Amortized Learning (CASAL), a method that enables large language models to answer questions they know while abstaining from those they do not. The proposed approach requires lightweight training, is claimed to be both efficient and effective, and is evaluated on language-only and vision-language benchmarks to reduce hallucination rates without significantly increasing unnecessary refusals. The authors also claim that this is the first steering-based training method shown to be effective for both dense and MoE/sparse setups.

**Strengths:**

* The paper is well written and easy to follow. It is also detailed and comprehensive.

* The experimental setup is fair and addresses several practical aspects of the solution, such as efficiency and preserving the original model’s capabilities on general tasks while reducing hallucinations. I particularly appreciate that the results are supported with confidence intervals.

* To assess the preservation of the original model’s general capabilities, the authors evaluated it on various setups, including general knowledge, reasoning, and coherence in multi-turn conversations. The results indicate that the models maintain overall performance comparable to the baseline.

* The proposed approach is evaluated in both the language and the vision-language domains, and architecture-wise covers LLMs, multimodal LLMs, and MoEs.

**Weaknesses:**

* It would be interesting to also compare against online RL algorithms in this setup, e.g., PPO or GRPO.

* In Tables 1 and 5, could you add reference performances from models like GPT-5 and Gemini 2 to provide a sense of the SOTA performance? Otherwise, the improvements are marginal in many cases with quite high standard deviations, so it is difficult to gauge if the improvements are significant.

* In fact, you can do a significance test to confirm whether the improvements are statistically significant or not, using the standard error at a 95% confidence level.

* It would be nice to evaluate the proposed method on at least two more LLMs of different sizes, such as Qwen 3, GPT-OSS, of varying sizes. Evaluating only on LLaMA 3.1 8B makes the setup weaker.

* For vision-language tasks, it would be nice to see how other baselines perform, e.g., DPO, SFT, or GRPO.

* WorldCuisines-VQA is not a hallucination-related dataset. Could you please discuss how you are testing hallucination with it?

* l.252: *“We compute Silhouette score as a measure of cluster separation.”* — Is this a standard technique in the related literature to understand the decision boundaries between known and unknown queries? If so, could you please share some references? Also, how does the decision boundary compare between DPO/SFT and CASAL?

* Based on Fig. 2, adding more data beyond 1280 does not lower the hallucination rate. Does this mean the method does not scale with more data? Is this because CASAL trains a lightweight one-layer model, which may saturate quickly?

* How are the standard deviations calculated in the tables?

* Also, when you are making the results bold, you should bold those numbers that are the best, considering the base model. If the base model is the best, then you should bold that instead of the applied methods. Otherwise, it is misleading the readers, see Table 1.

* In Table 1, there is a sharp drop (6%) in performance on POPQA -- can you explain why there is such a drop when applying CASAL? DPO seems to be performing better. In fact, you are deteriorating in both refusal rate and accuracy on POPQA.

* It seems the refusal rate increased and accuracy dropped after CASAL in OOD generalization, as per Tables 3 and 4. Can you comment on that? Also, it would be great to have some reference points here with DPO, GRPO, SFT, etc.

* What does “Before CASAL” refer to? Is it simply the base model? In tables where only “Before CASAL” and “After CASAL” are reported, I strongly recommend including additional reference points such as DPO, SFT, and GRPO. Otherwise, it is difficult to interpret the results. For example, in Table 5, training with WorldCuisine-QA using a specific loss compared to the base Qwen 2.5-VL model may show some improvement, but it does not convey much without comparisons to other loss functions.

* The appendix formatting seems to be problematic in some places. Please check p.46 (blank page) and p.47 (overflow).

**Questions:**

Please see weaknesses above; I remain open to adjusting my scores pending a convincing rebuttal addressing the identified weaknesses.

---

> ### Author Response · Authors · 2025-11-26
>
> ## 1.
> > “Is this a standard technique in the related literature to understand the decision boundaries between known and unknown queries? If so, could you please share some references?”
>
> Thank you for this question. The Silhouette score [1] (Rousseeuw, 1987) is a well-established clustering evaluation metric (cited by 26275) that evaluates cluster quality and separation by measuring both within-cluster cohesion and between-cluster separation. While its application to known/unknown query classification may be novel in our specific context, the underlying principle is well-founded in machine learning literature.
>
> It is particularly appropriate for our analysis because it quantifies both within-cluster cohesion and between-cluster separation—precisely the properties needed to assess whether known and unknown queries form distinguishable decision boundaries in the embedding space. To ensure reproducibility and clarity, we have included the formal equations and interpretation guidelines for the Silhouette score on page 36 of the revised manuscript.
>
> ---
> ## 2.
> > “Also, when you are making the results bold, you should bold those numbers that are the best, considering the base model. If the base model is the best, then you should bold that instead of the applied methods. Otherwise, it is misleading the readers, see Table 1”
>
> We have now considered the base model when bordering the results in the updated manuscript.
>
> ---
>
> ## 3.
> > “The appendix formatting seems to be problematic in some places. Please check p.46 (blank page) and p.47 (overflow).”
>
> We are grateful for the reviewer's detailed observation. We will fix the appendix formatting in the camera ready version of the manuscript.
>
> ---
>
> ## 4.
> > “How are the standard deviations calculated in the tables?”
>
> The standard deviations are calculated following the standard formula (using np.std package in code): $ \\sigma = \\sqrt{\\frac{\\sum_i (a_i - \\bar{a})^2}{N}}$
>
> ---
>
> ## References:
>
> [1] Peter J. Rousseeuw (1987). "Silhouettes: a Graphical Aid to the Interpretation and Validation of Cluster Analysis". Computational and Applied Mathematics. 20: 53–65. doi:10.1016/0377-0427(87)90125-7.

---

> ### Author Response · Authors · 2025-11-26
>
> We sincerely thank the reviewer for the positive appraisal and for recognizing our work as **'detailed and comprehensive.'** We particularly appreciate the acknowledgment of our rigorous experimental setup, specifically the inclusion of *confidence intervals* and the broad evaluation across architectures. The reviewer’s constructive suggestions have allowed us to refine the paper further and helped us further elevate the quality of this manuscript. We have conducted the following **new experiments and analysis** to address the main points raised by the reviewer:
>
> ## 1.
>
> We have added one more online RL learning baseline (GRPO) as suggested by the reviewer. As shown in Figure 2 of the revised manuscript, our conclusion remains consistent: CASAL is more compute efficient than all competitive baselines: SFT, DPO and GRPO.
>
> ---
>
> ## 2.
>
> We appreciate the reviewer’s recommendation to expand our baselines. We have systematically incorporated SFT, DPO, and GRPO into our experiments. The updated OOD generalization results are presented in Tables 3 and 4, while the multimodal results are detailed in Table 5.
>
> ---
>
> ## 3.
>
> We appreciate the reviewer's important clarification request regarding our muti-modal experiments – these are important results that demonstrate CASAL is modality-agnostic. We have now added an additional multi-modal dataset, "Landscape-VQA", to complement WorldCuisine-VQA. While WorldCuisine-VQA and Landscape-VQA are not traditionally hallucination benchmarks, we adapt them to evaluate hallucination via failure to abstain. Consistent with our text-based evaluation framework and with the literature [1], we define hallucination as a model confidently generating incorrect answers for unknown inputs instead of abstaining.
>
> * Our setup is:
>     * Dataset Partitioning: We use the knowledge probing step (Section 2.1) to partition each dataset into "known" vs. "unknown" image-question pairs based on the model's internal knowledge.
> Input: The VLM receives an image I (e.g., a rare French regional dessert) paired with a question Q (e.g., "What is this dish known as in France?").
>     * Hallucination Definition:
>         * Correct behavior (unknown image-question pair): Abstain → "I'm not certain about this specific dish"
>         * Hallucination (unknown image-question pair): Confident incorrect answer → "This is a tarte Tatin" (when it is not)
>
> This directly mirrors our text-based setup, where models must distinguish answerable from unanswerable queries. The only difference is the additional image modality. We have added specific examples, example model outputs, and expanded explanations and clarifications in the revised Table 5 and Appendix O (Page 48-50).
>
> ---
>
> ## 4.
>
> We thank the reviewer for proposing this insightful additional analysis. Comparing representations before and after learning across different training approaches proved to be very meaningful; however, due to space limitations, we have placed these new figures in the Supplementary Material (Appendix S, page 56).
> We performed PCA on the hidden layer activations to compare decision boundaries between DPO, SFT, GRPO, and CASAL. Consistent with our hypothesis, CASAL demonstrates the best cluster separation and the clearest boundary between known and unknown queries compared to the other methods. This validates that by directly training a local representation loss, CASAL effectively encourages a distinct separation between these activation states.
>
> ---
> ## Reference:
>
> [1] Ferrando et al. (2025). “Do I Know This Entity? Knowledge Awareness and Hallucinations in Language Models”. ICLR. https://arxiv.org/abs/2411.14257

---

### Author Response · Authors · 2025-12-03
**Summary of the Discussion Period**

Dear Area Chair and Reviewers,

 We wish to thank the Area Chair in advance for your upcoming time in considering our manuscript. We thank the reviewers for their constructive feedback and encouraging *initial* assessment (**Initial Ratings: 8, 6, 6, 6**).

The reviewers unanimously recognized the novelty of amortized activation steering, its efficiency, and broad applicability. We also want to summarise and highlight the central, novel contribution of our work:

- **Conceptual Novelty**: Our approach connects insights from two fields-- interpretability and amortized optimization. CASAL instantiates amortized optimization by incorporating activation steering into training: it "amortize" the activation steering process by training a lightweight subnetwork that learns to approximate the steering solution.

- **Fresh Paradigm**: CASAL is the first approach to rely solely on a representation-level objective for hallucination reduction in language models. Representation loss provides a learning signal from within: from model’s own hidden representation.

- **Efficiency**: A method that is 30$\times$ more compute-efficient and 20$\times$ more data-efficient than strong baselines.

- **Empirical Robustness**: Consistent hallucination reduction across diverse architectures (Dense, MoE) and modalities (Text, Vision) without degrading general capabilities.

---

# New Experiments and Clarifications:

During the rebuttal period, we conducted extensive new experiments to systematically address the primary concerns regarding baselines, hyperparameter sensitivity analysis and additional ablation studies. We believe these revisions resolve the few remaining minor concerns, solidifying the paper's contribution and even meriting for higher ratings.

1. Strengthened Baselines & Comparative Analysis

Reviewers CQ6A and rR6p requested stronger comparisons against on-line RL baselines, and additional baselines for multi-modal settings. We implemented and compared CASAL against GRPO across text and vision tasks, as well as generalisation setups. Updated results (Figure 2, Tables 3, 4, 5) demonstrate that CASAL remains significantly more compute-efficient while achieving superior or competitive hallucination reduction compared to these resource-intensive baselines.

---

2. Representation and Mechanism:

As requested by CQ6A and Th4B, we added PCA visualization (Appendix S) of hidden layer activations. We systematically compared the representation after  CASAL training with various baselines. This empirically demonstrate that CASAL creates a cleaner linear separation (decision boundary) between "known" and "unknown" states compared to DPO, SFT, or GRPO.

---

3. Expanded Multi-Modal Evaluation

Reviewer CQ6A asked about the relevance of WorldCuisines-VQA for hallucination.  We clarified our definition of VQA hallucination and provided qualitative examples. We also added a completely new dataset, Landscape-VQA, to the evaluation suite.

---

4. Comprehensive Ablation Studies & Sensitivity

Reviewers FqvX and rR6p asked for deeper analysis regarding hyperparameters (thresholds, layers, steering magnitude).

We added:
- Threshold Analysis: Determined $T_H=7$ as the optimal probing threshold for balancing data quality and vector accuracy (Figure 11).
- Layer Sensitivity: Confirmed middle layers provide the best trade-off between semantic abstraction and steerability (Appendix E).
- Steering Coefficient: Identified a stable "sweet spot" (magnitude 4–6) that works across architectures (Figure 16).
- Module Selection: Showed that while training the whole MLP is the best, specific sub-modules (up/down projections) are also effective (Section K).

---

5. Limitation Acknowledgement (Long-context, multi-turn and complex reasoning)

Reviewers noted the focus on short-form QA. We have acknowledged this boundary in the Limitations section. However, we cited recent literature (Obeso et al., 2025) which supports our hypothesis that internal representation steering is transferable to long-context and potentially reasoning tasks, marking this as a clear avenue for future work.

---

# Conclusion:

We have addressed all major concerns by adding additional baseline (GRPO), expanding the VLM dataset suite, and providing extensive ablation studies.

Given the method's proven efficiency and the reviewers' acknowledgment of its conceptual novelty ("fresh paradigm", "novel training framework") and "strong empirical results", we hope the AC finds this submission ready for acceptance.

We'd also appreciate the AC to consider the potential for score increases that might have occurred had the rebuttal period proceeded without the interruption caused by the reviewer/AC identity leak. For example, we'd like to highlight that reviewer CQ6A explicitly expressed that they "remain open to adjusting their scores pending a convincing rebuttal" in their **initial** review (*BEFORE* the incidents of leaked identity).

---

### Meta-Review · Area_Chair_fGT6 · 2026-01-05

**Summary:**

Overall this paper has a new and well-supported view for hallucination mitigation via representation-level method. Effectiveness and efficiency of the proposed method have been well justified. Reviewers think the method is novel and practical and that the method can be applied to LLMs and VLMs. The improvement is good. AC recommends accept based on the unanimous reviewer recommendations.

**Reviewer Concerns:**

The main concerns were about comparison with other methods, robustness and clarify of evaluation. Authors' rebuttal was good, where they added comparison to GRPO and multimodal methods. They also provided threshold, layer comparisons. They also clarified the multimodal hallucination definition with examples and an additional dataset.

**Reviewer Scores:**

I think reviewers will be generally happy with the rebuttal and improve or main the scores.

---

### Decision · Program_Chairs · 2026-01-26

Accept (Poster)